# Extracellular nanovesicles for packaging of CRISPR-Cas9 protein and sgRNA to induce therapeutic exon skipping

Peter Gee [1,2,3], Mandy S.Y. Lung [1], Yuya Okuzaki [1], Noriko Sasakawa[1], Takahiro Iguchi [1], Yukimasa Makita [3,4], Hiroyuki Hozumi[3,4], Yasutomo Miura[1], Lucy F. Yang[1], Mio Iwasaki [1], Xiou H. Wang [1], Matthew A. Waller [1], Nanako Shirai[1], Yasuko O. Abe [1], Yoko Fujita [5], Kei Watanabe [1], Akihiro Kagita[1], Kumiko A. Iwabuchi [1,3], Masahiko Yasuda[6], Huaigeng Xu [1], Takeshi Noda[5], Jun Komano [7,8], Hidetoshi Sakurai[1], Naoto Inukai [3,4] & Akitsu Hotta [1,2,3]✉

Prolonged expression of the CRISPR-Cas9 nuclease and gRNA from viral vectors may cause off-target mutagenesis and immunogenicity. Thus, a transient delivery system is needed for therapeutic genome editing applications. Here, we develop an extracellular nanovesicle-based ribonucleoprotein delivery system named NanoMEDIC by utilizing two distinct homing mechanisms. Chemical induced dimerization recruits Cas9 protein into extracellular nano-vesicles, and then a viral RNA packaging signal and two self-cleaving riboswitches tether and release sgRNA into nanovesicles. We demonstrate efficient genome editing in various hard-to-transfect cell types, including human induced pluripotent stem (iPS) cells, neurons, and myoblasts. NanoMEDIC also achieves over 90% exon skipping efficiencies in skeletal muscle cells derived from Duchenne muscular dystrophy (DMD) patient iPS cells. Finally, single intramuscular injection of NanoMEDIC induces permanent genomic exon skipping in a luciferase reporter mouse and in *mdx* mice, indicating its utility for in vivo genome editing therapy of DMD and beyond.

[1] Center for iPS Cell Research and Application (CiRA), Kyoto University, 53 Kawahara-cho, Shogoin, Sakyo-ku, Kyoto 606-8507, Japan. [2] Institute for Integrated Cell-Material Sciences (iCeMS), Kyoto University, Yoshida Ushinomiya-cho, Sakyo-ku, Kyoto 606-8507, Japan. [3] Takeda-CiRA Joint Program (T-CiRA), Fujisawa, Kanagawa, Japan. [4] T-CiRA Discovery, Takeda Pharmaceutical Company Limited, 26-1, Muraoka-Higashi 2-chome, Fujisawa, Kanagawa 251-8555, Japan. [5] Laboratory of Ultrastructural Virology, Institute for Frontier Life and Medical Sciences, Kyoto University, 53 Kawahara-cho, Shogoin, Sakyo-ku, Kyoto 606-8507, Japan. [6] Pathology Analysis Center, Central Institute for Experimental Animals, Kawasaki, Kanagawa 210-0821, Japan. [7] Department of Clinical Laboratory, Nagoya Medical Center, 1-1 4-chome, Sannomaru, Naka-ku, Nagoya 460-0001, Japan. [8] Present address: Department of Infection Control, Osaka University of Pharmaceutical Sciences, 4-20-1 Nasahara, Takatsuki, Osaka 569-1041, Japan ✉email: akitsu.hotta@cira.kyoto-u.ac.jp

Clustered regularly interspaced short palindromic repeat (CRISPR)-associated protein (Cas9) has enabled efficient editing of human cells in culture and has potential as a therapeutic tool for treating human diseases[1,2]. However, in vivo delivery of CRISPR-Cas9 is needed to target tissues of interest depending on the disease. Duchenne muscular dystrophy (DMD) is a severe muscle degenerative disease caused by mutations in the X-linked gene, dystrophin[3]. The absence of dystrophin protein in skeletal and cardiac muscle cells leads to a loss in muscle stability and results in muscle wasting[4]. CRISPR-Cas9 has been reported as an efficient tool for inducing exon skipping in iPSCs[5] and in vivo animal DMD models[6–9] to restore dystrophin protein expression.

Adeno-associated viruses (AAV) have been the leading tool for in vivo gene delivery, and utilized to treat DMD animal models by delivering the CRISPR-Cas9 system[6–9]. However, there are several limitations and concerns regarding its potential use in therapeutics including a limited viral genomic DNA packaging capacity (<5 kb)[10], neutralizing antibodies against AAV capsids[11,12], and immunogenicity to Cas9 protein[13]. Moreover, prolonged expression of a transgene by AAV can be observed for several years[14]. For SpCas9 expression, this would not be ideal as it might result in unwanted off-target mutagenesis[15–17]. Indeed, a recent report demonstrated that mice treated with AAV vector delivering SaCas9 showed immunogenicity as well as integration of the AAV vector DNA fragments into the host genome[18]. In order to minimize these adverse effects, a transient delivery method is desired.

Ribonucleoprotein (RNP) delivery of CRISPR-Cas9 offers several advantages over DNA delivery[19]. It facilitates potent on-target cleavage while also reducing unwanted off-target effects, as RNP is rapidly degraded in cultured cells compared with DNA plasmid expression vectors[20]. However, delivering CRISPR RNP complexes into hard-to-transduce tissues requires a suitable delivery system that can efficiently package, protect and deliver cargo of interest into target tissues.

Viruses are natural carriers of proteins and nucleic acids for delivery into cells. Created by expressing viral envelope and/or structural proteins, extracellular vesicles (EVs) lacking a viral nucleic acid genome can be harnessed into protein and RNA delivery vehicles, and have been utilized for clinical trials in the vaccination field[21]. The structural polyprotein from retroviruses, Gag, is an ideal candidate to package cargo into EVs as it is well studied and can induce active release of EVs from cells, which have been estimated to contain up to 5000 Gag molecules per virus particle[22]. Previously, we and others have fused HIV and MLV Gag with proteins of interest for delivery into cells[23–27].

Although EV-mediated CRISPR-Cas9 RNP delivery methods have been reported, such as Cas9P LV[28] and NanoBlades[29] systems that fuse SpCas9 with retroviral Gag, VEsiCas[30] system that passively incorporates SpCas9, or Gesicle[31] system that uses dimerization based incorporation of SpCas9, applicability for in vivo muscle tissue has not been elucidated. Furthermore, direct fusion of SpCas9 with Gag requires the supplementation of wildtype Gag-Pol to liberate Cas9 from Gag via protease-mediated cleavage, which competes for space within the EV and reduces the number of SpCas9 molecules packaged. Inclusion of protease in Pol also runs the risk of protease-mediated degradation at cryptic sites in the target protein, inadvertently reducing the number of functional proteins delivered[23]. Thus, there is a need for active incorporation machinery for Cas9 protein and sgRNA, which does not involve direct fusion of Cas9 protein with Gag.

Here, we develop an all-in-one EV delivery system termed NanoMEDIC (nanomembrane-derived extracellular vesicles for the delivery of macromolecular cargo). NanoMEDIC efficiently induces genome editing in various human cell types, such as T cells, monocytes, iPSCs, iPSC-derived cortical neurons, and myogenic cells. NanoMEDIC can also be multiplexed to simultaneously target the splicing acceptor (SA) and donor sites (SD) of DMD patient iPSCs and in a transgenic luciferase reporter mouse.

## Results

**Ligand-induced EV packaging system with HIV Gag and SpCas9.** We sought to develop a suitable chemical-induced dimerization system for the incorporation of SpCas9 protein into EVs in producer cells, which would be advantageous for SpCas9 release and translocation into the nucleus of target cells. We chose the FKBP12 and FRB dimerization system, which has been extensively used for protein translocation studies[32]. An FRB variant (T2098L) specifically binds to rapamycin analog, AP21967, with high affinity[33]. We repurposed this interaction pair for selectively packaging SpCas9 protein into budding EVs from producer cells. Initially, three membrane-anchoring proteins were assessed for their ability to incorporate N-terminal fused FRB-SpCas9 into EVs, namely VSV-G-FKBP12, LM-FKBP12-Gag containing the myristoylation motif from human Lyn kinase (LM)[23], and LM-FKBP12-EGFP (Fig. 1a). VSV-G-FKBP12 was chosen as a candidate because VSV-G is an envelope glycoprotein derived from vesicular stomatitis virus and is commonly used to pseudotype lentivirus and retrovirus vectors for its broad tropism. LM-FKBP12-EGFP was selected as a candidate because of its ability to target the plasma membrane of cells and potential to be passively packaged into budding EVs. It is worth noting that both VSV-G-FKBP12 and LM-FKBP-EGFP are expected to be passively packaged into budding EVs. On the other hand, HIV Gag was chosen as a candidate as it has been previously reported to deliver proteins of interest by direct fusion.

Focusing on SpCas9 protein delivery, EVs were produced in the absence or presence of AP21967, and then inoculated onto HEK293T cells stably expressing sgRNA DMD1 (Fig. 1b), which targets the SA site of exon 45 in the human dystrophin gene, herein labeled as sgRNA-DMD1[5]. Incorporated SpCas9 protein was visualized by western blot analysis of EVs (Supplementary Fig. 1A). Subsequently, genomic indels of the target cells were observed by T7E1 assay. FKBP12-Gag packaged SpCas9 more efficiently than the other two membrane-anchoring proteins in the presence of AP21967, which led to higher genomic DNA editing activity when delivered into target HEK293T cells stably expressing sgRNA-DMD1 (Fig. 1c). Hence, we selected this construct for further experiments.

We next optimized the position of FRB fused with SpCas9 at the N-terminus, C-terminus, or N- and C-terminus. FRB fusion protein activity was compared with WT SpCas9 in HEK293T cells transiently transfected with the fusion construct expression plasmids together with a plasmid encoding sgRNA-DMD1 (Supplementary Fig. 1B). The activity of all fusion proteins was comparable with WT SpCas9 except for the N- and C-terminus FRB-fused SpCas9, which had lower expression in producer HEK293T cells (Supplementary Fig. 1C). We next generated and inoculated the EVs onto HEK293T cells stably carrying a single-strand annealing (SSA) EGFP reporter (EGxxFP), where the GFP coding region is interrupted by a 100 bp sequence containing the sgRNA-DMD1 target sequence (Fig. 1d). Upon targeted DNA cleavage, single-strand annealing occurs and EGFP + expression is restored. N-terminal fused SpCas9 had the highest packaging efficiency into EVs and delivery into reporter cells compared with two other constructs in the presence of AP21967 (Fig. 1e), even though fusion proteins were packaged at similar levels in the EVs (Supplementary Fig. 1D). These results indicate that FRB N-terminal fused SpCas9 may dissociate from EVs more efficiently in target cells.

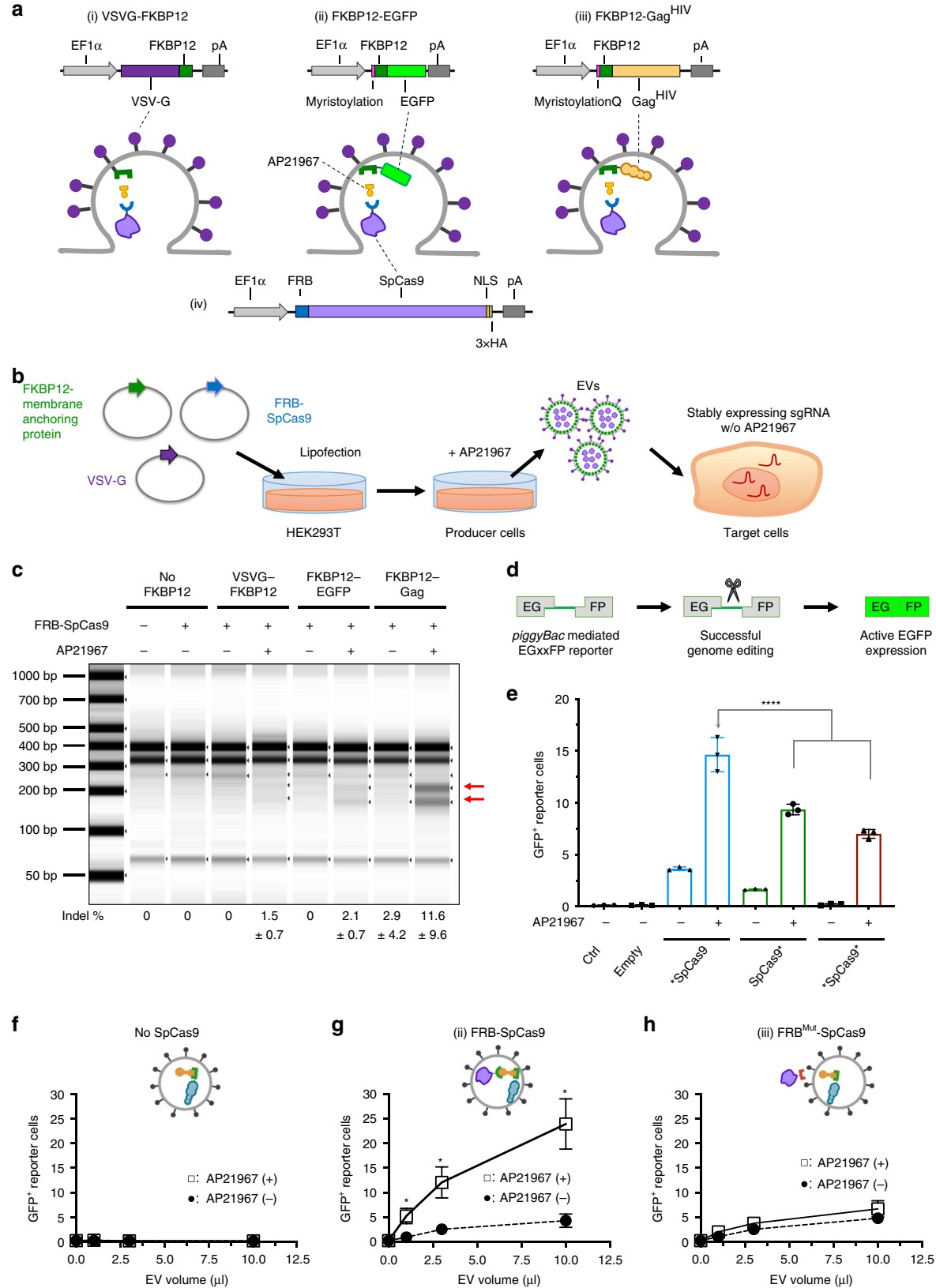

To confirm the specificity of ligand-dependent dimerization of FRB, leucine at amino-acid position 2098 was mutated to alanine (FRB[Mut]), as it is critical for AP21967-induced dimerization[33]. This mutation abrogated SpCas9 recruitment into EVs in the presence of AP21967, indicating that ligand-dependent Cas9 incorporation was owing to the specific interaction between FRB and FKBP12, rather than passive incorporation (Fig. 1f–h). Hereafter, we term our chemical-induced dimerization EV system as NanoMEDIC.

**Fig. 1 Selective packaging of SpCas9 protein into NanoMEDIC by chemical-induced dimerization. a** Schematics of membrane-anchoring constructs fused with FKBP12 and FRB-SpCas9. **b** Schematic of NanoMEDIC production from producer cells and delivery into recipient cells. EGFP is neon green. The myristoylation domain is indicated in pink. The FKBP12 heterodimerization domain is dark green and the FRB heterodimerization domain is blue. Gag[HIV] is tan. SpCas9 protein is light purple. The dimerization ligand, AP21967, is yellow. VSV-G envelope is dark purple and on the surface of the cell. **c** T7E1 analysis of HEK293T cells stably expressing sgRNA targeting DMD1. These cells were inoculated with NanoMEDIC containing FRB-SpCas9 and no FKBP12 interaction partner, VSVG-FKBP12, FKBP12-EGFP-A, or FKBP12-Gag[HIV], produced in the presence or absence of AP21967. Red arrowheads show cleaved products by T7E1enzyme. Data are mean ± S.D. from technical triplicates. **d** Schematic showing NanoMEDIC inoculation onto HEK293T EGxxFP reporter cells stably expressing DMD1-sgRNA and the resulting GFP expression upon cleavage by delivered SpCas9 protein. **e** HEK293T EGxxFP reporter cells stably expressing sgRNA were inoculated with NanoMEDIC containing FKBP12-Gag[HIV] and FRB-SpCas9 (*SpCas9), SpCas9-FRB (SpCas9*), or FRB-SpCas9-FRB (*SpCas9*). The asterisks indicate the position of the FRB dimerization domain on SpCas9. ****, $P < 0.0001$ compared with *SpCas9 by one-way ANOVA. Mean ± S.D. from technical triplicates. **f–h** HEK293T EGxxFP reporter cells stably expressing sgRNA-SA were inoculated with increasing volumes of NanoMEDIC particles produced **f** without FRB-SpCas9, **g** with FRB-SpCas9, or **h** with FRB[Mut]-SpCas9 in the presence (+) or absence (−) of AP21967. *, $P < 0.01$ by multiple $t$ tests. $P$ values for 1, 3, and 10 μl were calculated to be 0.009, 0.007, and 0.003, respectively. Mean ± S.D. from technical triplicates. Source data are provided as a Source Data file.

**Packaging signal loading of sgRNA and ribozyme release**. Typically, sgRNA expression is mediated by an RNA polymerase III promoter (i.e., U6 promoter) and reported to localize in the nucleus[34]. However, for EV loading, sgRNA should be exported into the cytoplasm and localized near budding EVs for successful packaging in producer cells. To specifically incorporate sgRNA into NanoMEDIC particles, we constructed an expression vector with two lentiviral vector components, the Tat activation response element (TAR) in the 5′ LTR promoter region and an extended Psi ($\Psi^+$) packaging signal that binds specifically with nucleocapsid of Gag[35], to express mRNA containing an AmCyan-coding sequence. We reasoned that Tat could boost full-length RNA expression from the 5′-LTR promoter and the $\Psi^+$ packaging signal could direct RNA incorporation more efficiently than stochastic incorporation. Furthermore, to release packaged mRNA from NanoMEDIC after inoculation into target cells, we flanked sgRNA by self-cleaving ribozymes[36,37], HH and HDV ribozymes, and inserted them between the $\Psi^+$ and AmCyan cDNA. Ribozyme-mediated self-cleavage would liberate the sgRNA from long mRNA (Fig. 2a). To test the effect of $\Psi^+$-mediated packaging, we removed the $\Psi^+$ packaging signal from 5LTR-Psi-RGR to make 5LTR-ΔPsi-RGR (Fig. 2a).

NanoMEDIC loaded with SpCas9 were co-packaged with U6-sgRNA, 5LTR-Psi-RGR, or 5LTR-ΔPsi-RGR in the presence or absence of Tat. Resulting NanoMEDIC particles were inoculated onto EGxxFP SSA reporter HEK293T cells (without sgRNA expression) and flow cytometry was performed 3 days post inoculation. As shown in Fig. 2b, the editing efficiency of NanoMEDIC was threefold higher when sgRNA was packaged with 5LTR-Psi-RGR in the presence of Tat (lane 4) than U6-sgRNA (lane 2). Interestingly, when $\Psi^+$ was deleted, the functional delivery of gRNA was abrogated indicating that $\Psi^+$ is essential for specific incorporation into NanoMEDIC (Fig. 2b, lane 6). Tat was also essential for packaging of gRNA into EVs (Fig. 2b, lane 3).

**HIV protease attenuates functional SpCas9 protein**. We investigated the effect of HIV protease (Pol) on SpCas9 protein incorporation in SSA-GFP reporter cells. FKBP12-Gag showed higher editing efficiency than FKBP12-Gag-Pol, suggesting the elimination of Pol is advantageous for SpCas9 delivery (Supplementary Fig. 2A). Adding Darunavir, a clinically approved HIV-1 protease inhibitor, during NanoMEDIC production using FKBP12-Gag-pol increased the delivery of functional SpCas9 protein (lane 5), when compared with no Darunavir (lane 4). FKBP12-Gag-pol with Darunavir (lane 5) was almost as functional as NanoMEDIC produced with FKBP12-Gag (Supplementary Fig. 2A, lane 6), which lacks protease (Supplementary Fig. 2B). By western blot analysis of NanoMEDIC particles produced with FKBP12-Gag-pol, SpCas9 was cleaved in the absence

of protease inhibitor Darunavir, suggesting that the increased activity was due to a higher amount of full-length SpCas9 protein (Supplementary Fig. 2C).

**sgRNA and AP21967 synergistically recruit Cas9 RNP**. To evaluate the contribution of RGR (ribozyme-sgRNA-ribozyme) sgRNA packaging to recruit active SpCas9 complexes into particles, we generated NanoMEDIC in the presence or absence of AP21967 and RGR sgRNA, and inoculated them onto HEK293T EGxxFP reporter cells with or without stably expressing sgRNA-DMD1 (Fig. 2c). In reporter cells lacking sgRNA, only NanoMEDIC produced in the presence of RGR-DMD1 induced genome editing (lanes 5 and 6), and NanoMEDIC generated with AP21967 induced higher GFP positive cells (lane 4) than without AP21967 (Fig. 2c, lane 3). In sgRNA stably expressing reporter cells, SpCas9 protein incorporation appeared to be facilitated by AP21967 treatment (lane 4) or RGR packaging (lane 5). This result suggests that RGR mRNA may interact with FKBP12-Gag through $\Psi^+$ signal and FRB-SpCas9 through sgRNA scaffold. Importantly, SpCas9 incorporation was synergistically increased by RGR packaging and AP21967-mediated dimerization (Fig. 2c, lane 6). Western blot analysis of NanoMEDIC also confirmed that SpCas9 protein incorporated into the particles was maximal in +AP21967/+RGR NanoMEDIC samples (Supplementary Fig. 3A). Finally, all-in-one NanoMEDIC containing (+) sgRNA induced SSA-GFP+ expression in HEK293T EGxxFP reporter cells in a dose-dependent manner, whereas (−) sgRNA NanoMEDIC had no activity (Fig. S3B).

Proteome analysis of NanoMEDIC particles with exosomes obtained from HEK293T cells (Supplementary Fig. 3C) revealed components of NanoMEDIC such as SpCas9, GagHIV, and VSV-G were highly enriched while Tat and AmCyan expressed during the production of NanoMEDIC were present in low quantities (Supplementary Fig. 3C). Common exosome markers (CD63 and CD81) and known Gag associated host factors (IGF2BP1 and PPIA) were also enriched in NanoMEDIC particles.

**Comparison with other dimerization based RNP delivery system**. We compared NanoMEDIC against the CRISPR-Cas9 Gesicle Production system[31], which also utilizes chemical-induced heterodimerization to package SpCas9 but relies on a U6 promoter-driven sgRNA expression vector for packaging of sgRNA and a cherry picker membrane-anchoring protein as opposed to Gag[HIV]. As shown in Fig. S3D in HEK293T EGxxFP reporter cells, NanoMEDIC induced a significantly higher amount of GFP+ reporter cells. These results suggest that chemical ligand-induced dimerization in combination with our sgRNA packaging approach leads to more efficient all-in-one EVs than previously reported.

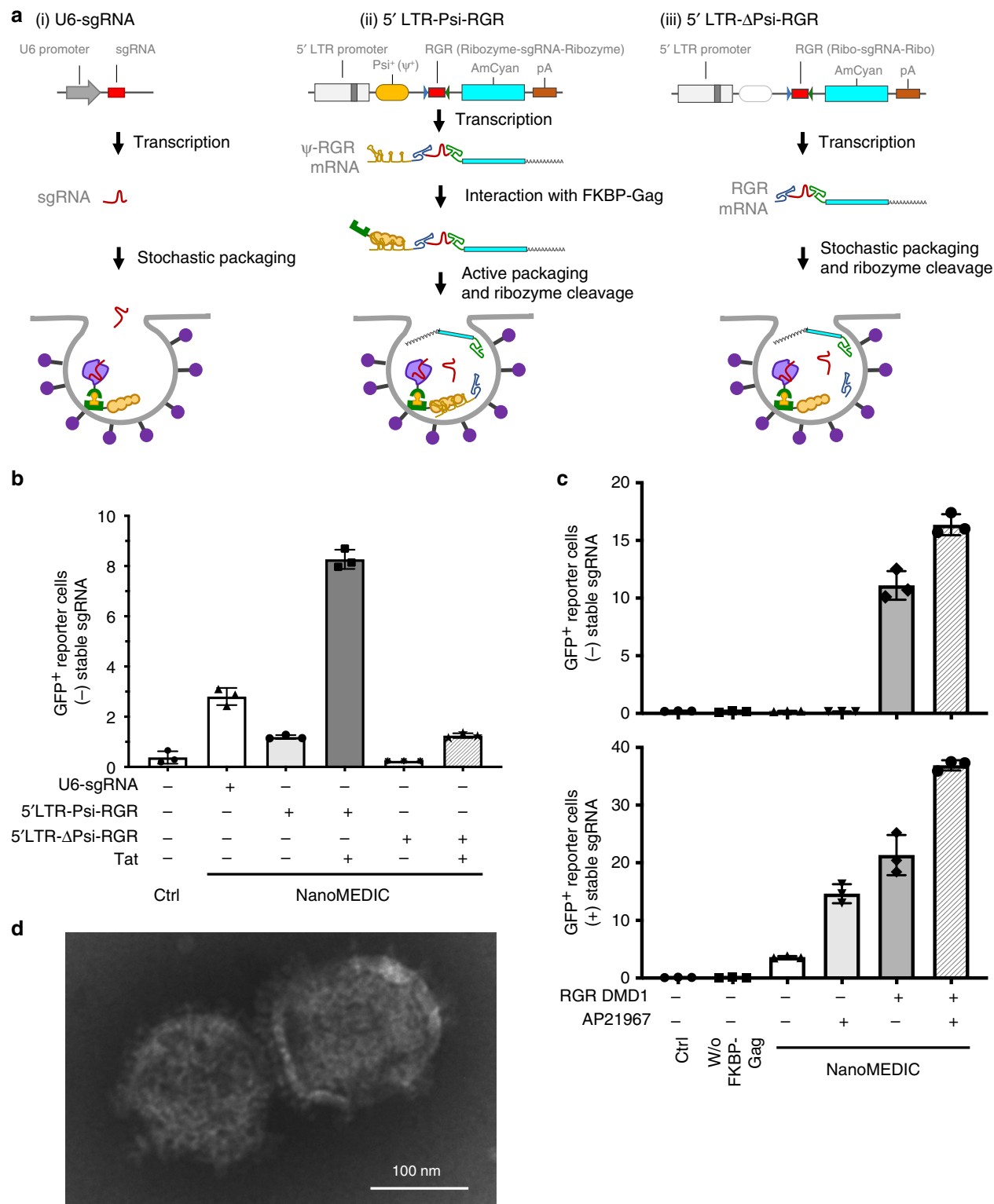

**Characterization of NanoMEDIC**. To further characterize NanoMEDIC particles, we purified NanoMEDIC by quaternary amine-based affinity chromatography. NanoMEDIC particles eluted at 0.65 M and 1 M NaCl (Supplementary Fig. 4A) were subjected to electron microscopy analysis (Fig. 2d). Spherical NanoMEDIC particles with an average diameter of 132 nm and 144 nm were observed in 0.65 M and 1 M purified NanoMEDIC particles, respectively (Supplementary Fig. 4B).

To determine nanoparticle size distribution and number, we performed light scattering and Brownian motion-based nanoparticle tracking analysis utilizing the NanoSight system. Analysis revealed a similar average diameter of 155 nm and 162 nm for the 0.65 M and 1 M purified NanoMEDIC particles, respectively (Supplementary Fig. 4C). Furthermore, the particle number was determined to be $3 \times 10^{12}$ and $6 \times 10^{12}$ particles per mL for the 0.65 M and 1 M elution samples, respectively. By performing an

**Fig. 2 HIV Tat and Ψ+ packaging signal are necessary for selective packaging of sgRNA into NanoMEDIC. a** Schematics of sgRNA expression vectors: (i) U6-sgRNA, sgRNA is stochastically incorporated into particles; (ii) 5LTR-Psi-RGR, sgRNA is actively packaged into budding particles through an interaction of Psi+ (Ψ+) with Gag, after which ribozymes self-cleave to liberate the sgRNA; (iii) 5LTR-ΔPsi-RGR, without the Ψ+ packaging signal, RNA is stochastically packaged into particles, after which ribozymes self-cleave. hU6: human U6 (Pol III) promoter; 5LTR: 5′ long terminal repeat (Pol II) promoter; Ψ+: extended packaging signal; RGR: hammerhead (HH) ribozyme, sgRNA, and hepatitis delta virus (HDV) ribozyme. **b** Tat and Ψ+ increased sgRNA packaging into NanoMEDIC. HEK293T EGxxFP reporter cells were inoculated with NanoMEDIC produced with different sgRNA expression vectors in the presence or absence of Tat. GFP reporter expression was analyzed by flow cytometry analysis 3 days after inoculation. Mean ± S.D. from technical triplicates. **c** AP21967 and sgRNA expression synergistically recruit FRB-SpCas9 into NanoMEDIC. Upper panel: HEK293T EGxxFP reporter cells were inoculated with NanoMEDIC particles that were produced with different combinations of plasmids expressing RGR-DMD1 and AP21967. Lower panel: HEK293T EGxxFP reporter cells stably expressing sgRNA were inoculated with the same amount NanoMEDIC particles. Flow cytometry analysis was performed 3 days after inoculation. Mean ± S.D. from technical triplicates. **d** Transmission electron microscopic analysis of purified NanoMEDIC particles revealed spherical structure with 130–140 nm in diameter. Results are representative of 27 electron microscopy images. Source data are provided as a Source Data file.

in vitro cleavage assay to quantify the amount of active RNP complex in NanoMEDIC particles compared with a standard RNP activity curve (Supplementary Fig. 4D), we calculated the number of active RNP molecules per particle was 3.5 and 7.9 in the 0.65 M and 1 M elution samples, respectively (Supplementary Fig. 4E).

**Screening of sgRNA for DMD exon skipping**. To achieve efficient exon skipping activity, potent sgRNA is critical. Dystrophin exon 45 is the second most common target next to exon 51[5] for DMD exon skipping. For facilitating sgRNA screening, we constructed a luciferase-based reporter system for detecting dystrophin exon 45 skipping activity. Firefly luciferase2 cDNA was split by inserting human dystrophin exon 45 flanked by intronic sequences (~ 700 bp) (Fig. 3a). We introduced a point mutation in the luciferase cDNA (c.G967A, p.V323I) to avoid a cryptic SD site for reducing background luciferase activity. By transfecting plasmid DNAs into HEK293T cells, we tested 26 sgRNAs with NGG PAM that target human exon 45 and picked the top six sgRNAs. We hypothesized that disruption of spliceosomes can be more efficient if two splicing regulatory sites are simultaneously disrupted. The combination of two sgRNAs, sgRNA-DMD1 and -DMD23, which target SA and SD sites, respectively, had the highest exon skipping activity (> 50-fold, Fig. 3c). Therefore, we generated two types of NanoMEDIC, each containing sgRNA DMD1 and DMD23, respectively, for testing simultaneous delivery of multiple particles into a single cell.

**NanoMEDIC efficiently edits various cells**. To show that NanoMEDIC can edit a variety of cell types, we inoculated NanoMEDIC onto differentiated C2C12 mouse myotubes and undifferentiated human Hu5 myoblasts stably integrated with the same EGxxFP SSA reporter (with DMD1 target site but no DMD23 site) that we used in HEK293T cells. NanoMEDIC packaged with sgRNA-DMD1-induced SSA-GFP + expression in the differentiated C2C12 myofibers and Hu5 cells, in contrast to no induction by NanoMEDIC with non-targeting sgRNA-DMD23 (Fig. 3d–f). NanoMEDIC efficiently induced indels in human 404C2-induced pluripotent stem cells (iPSCs) in a dose-dependent manner (Fig. 3g). In 404C2 and 1383D2 healthy iPSCs, multiplex delivery of two types of NanoMEDIC particles, one targeting the SA site (DMD1) and the other targeting the SD (DMD23) site, resulted in exon 45 deletion up to 22% and 29%, respectively (Fig. 3h).

To expand the utility of NanoMEDIC, we tested additional gene loci in various human cell types. Targeting the CCR5 gene (HIV co-receptor) in Jurkat T-lymphocyte cells, NanoMEDIC induced a high degree of indels in a dose-dependent manner up to 48% by T7E1 analysis (Supplementary Fig. 5A). NanoMEDIC could also knockout EGFP expression in U937 monocyte cells

stably expressing EGFP (Supplementary Fig. 5B). Moreover, NanoMEDIC delivered onto iPSC-derived cortical neurons could efficiently edit the SAMHD1 gene, which has implications in congenital encephalopathy, using two sgRNAs with efficiencies up to 36% (Supplementary Fig. 5C).

To demonstrate reproducibility, we generated six NanoMEDIC batches targeting six genomic loci and quantified the amount of active RNP complex for each batch in vitro. Subsequently, we inoculated NanoMEDIC containing 50, 200, or 500 ng of equivalent active RNP complex onto HEK293T cells, in a side-by-side comparison with plasmid DNA transfection to express Cas9 and corresponding sgRNA. NanoMEDIC outperformed plasmid DNA transfection in all six endogenous genomic loci we tested (Supplementary Fig. 5D).

**Exon skipping and dystrophin expression in DMD patient iPSCs**. As a proof of concept, NanoMEDIC was targeted against the dystrophin gene of DMD patient iPSCs. CRISPR SpCas9 targeting of the exon 45 SA site can induce exon 45 skipping and restore dystrophin protein expression in iPSC-derived skeletal muscle cells[5]. We extended this strategy to target the SD site of exon 45 to enhance skipping. We produced two types of Nano-MEDIC containing sgRNA targeting the SA site (DMD1) or SD site (DMD23) and treated three different iPSC cell lines, two DMD patient lines lacking exon 44 and 46–47, respectively, and a healthy iPSC line. High indel frequencies over 50% were observed at SA and SD sites, respectively (Fig. 4a). When NanoMEDIC was multiplexed, no obvious inhibitory effect was observed and up to 38% of exon 45 was deleted (Fig. 4b, c).

Next, we differentiated Δexon 44 DMD iPSCs into skeletal muscle cells by MYOD1 overexpression as previously reported[38] to analyze exon skipping efficiency. NanoMEDIC targeting SA induced up to 36% exon 45 skipping, while SD targeting NanoMEDIC alone had weak exon skipping activity (Fig. 4d). Interestingly, when SA and SD NanoMEDIC were multiplexed, up to 92% exon skipping could be achieved, indicating a synergistic effect by both sgRNAs (Fig. 4d). Dystrophin protein expression in the iPSC-differentiated skeletal muscle cells correlated with exon skipping data and were highest when NanoMEDIC targeting SA and SD were multiplexed (Fig. 4e).

**Cell viability is not affected by NanoMEDIC delivery**. To test cell toxicity of NanoMEDIC treatment, HEK293T EGxxFP reporter cells were inoculated with NanoMEDIC produced without sgRNA or with RGR-DMD1. As observed before, RGR-DMD1 containing NanoMEDIC efficiently induced SSA-GFP cleavage in the reporter cells (Fig. 5a). However, the specific and transient DNA cleavage activity did not affect the cell proliferation when compared with non-treated cells and cells treated with Nano-MEDIC lacking sgRNA 48 hours post inoculation (Fig. 5b).

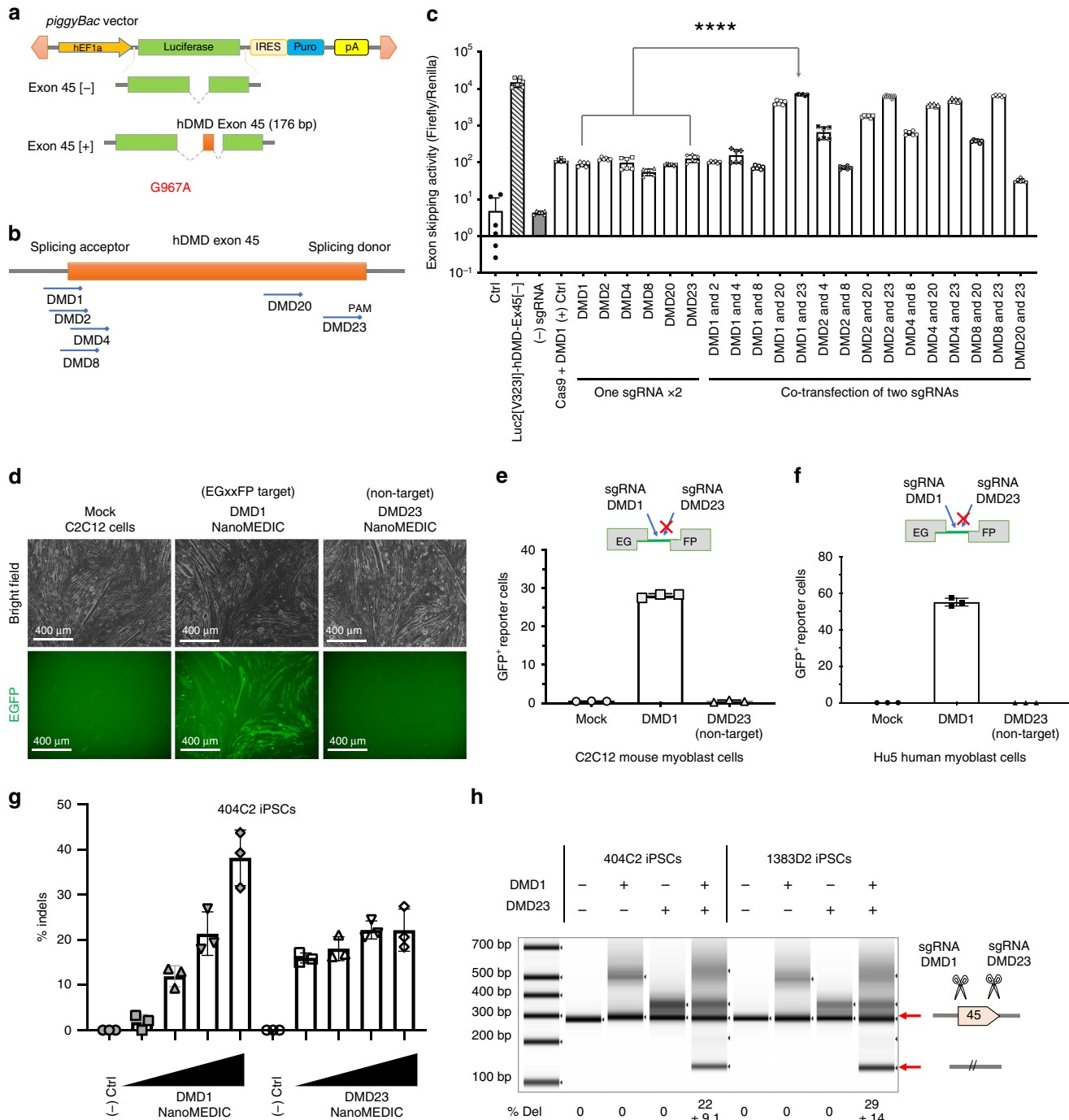

**Fig. 3 sgRNA screening to target the splicing acceptor and splicing donor sites of dystrophin exon 45. a** Schematic of *piggyBac* vector containing the luciferase reporter interrupted by human dystrophin exon 45 flanked by intronic regions. **b** Schematic representation of the most active sgRNAs. **c** Luciferase exon skipping reporter activity in HEK293T cells comparing single sgRNA and dual sgRNAs transfected together with SpCas9 plasmid. Exon skipping activity: mean ± S.D. from two experiments performed in technical triplicates. ****, $P < 0.0001$ by one-way ANOVA. **d, e** C2C12 EGxxFP cells were differentiated into mature myoblasts and inoculated with NanoMEDIC targeting either the SA or SD site of exon 45. Only the DMD1-targeting sequence is contained in the reporter, not the DMD23-targeting sequence, which is indicated with an X. **d** Bright field and GFP images that took 4 days after inoculation. **e** SSA-GFP+ expression analysis by flow cytometry. The results are depicted in the bar graph. Only the DMD1-targeting sequence is contained in the reporter, not the DMD23-targeting sequence, which is indicated with an X. Percent indels: mean ± S.D. from technical triplicates. **f** Hu5 EGxxFP human myoblasts were inoculated with NanoMEDIC containing RGR-DMD1 or RGR-DMD23.. Only the DMD1-targeting sequence is contained in the reporter, not the DMD23-targeting sequence, which is indicated with an X. Percent indels: mean ± S.D. from technical triplicates. **g** 404C2 iPSCs were inoculated with increasing concentrations of NanoMEDIC containing RGR-DMD1 or RGR-DMD23. T7E1 analysis was performed to measure the indel percentage, which increased in a dose-dependent manner with increasing amounts of NanoMEDIC (1 ng, 3 ng, 10 ng, and 30 ng of active RNP complex). Cleavage products are indicated by the red arrows. Percent indels: mean ± S.D. from technical triplicates. **h** Multiplexing NanoMEDIC produced with RGR-DMD1 and RGR-DMD23, respectively, resulted in dystrophin exon 45 deletion, measured by PCR amplification of genomic DNA of iPSCs from two healthy donors. iPSCs were treated with either RGR-DMD1, RGR-DMD23, or RGR-DMD1+DMD23 NanoMEDIC. PCR amplification of dystrophin exon 45 deleted genomic DNA is indicated with a red arrow. Percent of deletion: mean ± S.D. from technical triplicates. Source data are provided as a Source Data file.

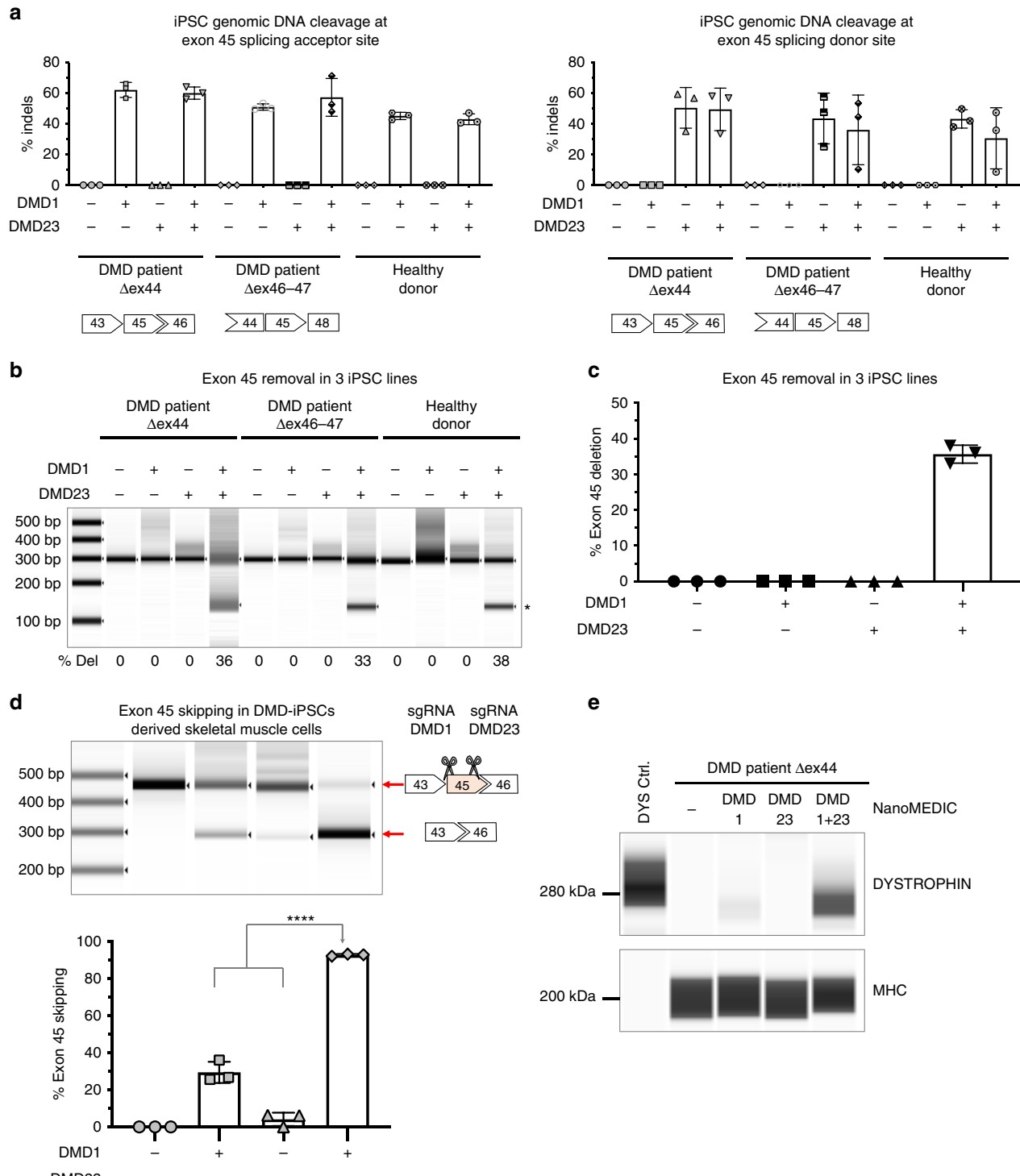

**Fig. 4 NanoMEDIC induces highly efficient genome editing in DMD patient iPSCs. a** iPSCs of a healthy donor, Δex44 DMD iPSCs, and Δex46-47 DMD patient iPSCs were inoculated with NanoMEDIC containing RGR-DMD1 (266 ng active RNP complex), RGR-DMD23 (190 ng active RNP complex), or RGR-DMD1 + RGR-DMD23 (266 and 190 ng active RNP complex, respectively). Three days after inoculation, the indel % was analyzed by T7E1 assay. Indel % is depicted at either the SA or SD site of *dystrophin* exon 45. % Indels: mean ± S.D. from technical triplicates. **b** Exon 45 removal from gDNA was determined by PCR amplification. The lower band with the asterisk indicates exon 45 removal by simultaneous cleavage by RGR-DMD1 and RGR-DMD23 NanoMEDIC. **c** Bar graph representation of average exon 45 removal as from three difference iPSC lines, iPSCs of a healthy donor, Δex44 DMD iPSCs, and Δex46-47 DMD patients shown in **b**. Mean ± S.D. from three experiments. % Deletion: mean ± S.D. from three independent cell lines. **d** DMD patient-derived Δex44 iPSCs, treated with RGR-DMD1, RGR-DMD23 or RGR-DMD1 + RGR-DMD23 NanoMEDIC, were differentiated into skeletal muscle cells. PCR amplification of exons 43 to 46 from extracted cDNA was performed. The lower band indicates exon 45 skipping. Mean ± S.D. from technical triplicates. ****, $P < 0.0001$ by one-way ANOVA. **e** Dystrophin protein expression was also analyzed by ProteinSimple Wes and compared with HEK293T cells overexpressing dystrophin cDNA. Myosin heavy chain protein was also analyzed as a loading control. Results are representative of two independent Protein Wes runs from a single experiment. Source data are provided as a Source Data file.

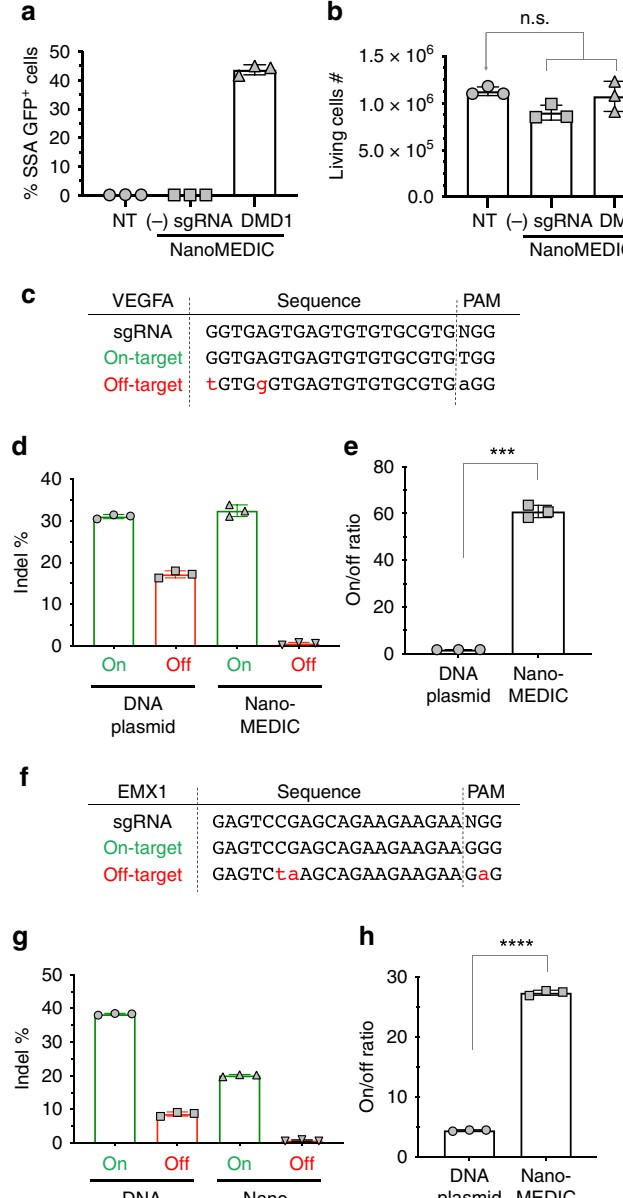

**Fig. 5 NanoMEDIC-mediated gene editing is not toxic and reduces off-target activity compared with plasmid DNA transfection. a, b** All-in-one NanoMEDIC particles containing RGR-DMD1 or no sgRNA were inoculated onto HEK293T EGxxFP reporter cells. SSA-GFP + expression was analyzed by flow cytometry **a** and cell number was counted 2 days after inoculation **b**. n.s. not significant by one-way ANOVA. % SSA–GFP + cells and living cell #: mean ± S.D. from technical triplicates. **c–h** HEK293T cells were either transfected with CRISPR DNA plasmids or inoculated with NanoMEDIC targeting VEGFA or EMX1. Three days after treatment of the cells, genomic DNA was extracted and indel % was determined by TIDE analysis. Mean ± S.D. from technical triplicates. **c** Sequences for VEGFA sgRNA, and on-target and off-target sites are shown. **d** Indel % at the VEGFA on-target and off-target sites are shown for DNA plasmid and NanoMEDIC delivery. Mean ± S.D. from technical triplicates. **e** The on/off ratio using each delivery method at the VEGFA site is depicted. **, $P = 0.005$ by unpaired two-tailed $t$ test. Mean ± S.D. from technical triplicates. **f** Sequences for EMX1 sgRNA, and on-target and off-target sites are shown. **g** Indel % at the EMX1 on-target and off-target sites are shown for DNA plasmid and NanoMEDIC delivery. Mean ± S.D. from technical triplicates. **h** The on/off ratio using each delivery method at the EMX1 site is depicted. ****, $P < 0.0001$ by unpaired two-tailed $t$ test. Mean ± S.D. from technical triplicates. Source data are provided as a Source Data file.

**Off-target analysis of NanoMEDIC with promiscuous sgRNAs.** We investigated off-target cleavage activity via NanoMEDIC, as it is a major concern for CRISPR-Cas9 clinical application. We compared off-target cleavage activities of NanoMEDIC and plasmid DNA delivery, utilizing previously reported promiscuous sgRNAs targeting VEGFA and EMX1 loci in HEK293T cells. On-target cleavage activity of VEGFA with NanoMEDIC was 32.5% and comparable with 31.5% indels by DNA plasmid delivery (Fig. 5c, d). Importantly, off-target cleavage activity was nearly eliminated using NanoMEDIC compared with DNA plasmid delivery. Furthermore, the on- to off-target ratio was over 70-fold for NanoMEDIC versus 1.8-fold for DNA plasmid (Fig. 5e). We found a similarly high on- to off-target ratio with EMX1 Nano-MEDIC of 27-fold versus 4.1-fold for DNA plasmid transfection (Fig. 5f–h). These results are in line with previous reports that transient delivery of CRISPR-Cas9 RNP induces lower off-target cleavage[20].

**NanoMEDIC-mediated protein delivery of luciferase in vivo.** To visualize protein delivery in vivo, we developed NanoMEDIC-Luc containing FRB-fused luciferase protein (Supplementary Fig. 6A). We confirmed luciferase protein enrichment in NanoMEDIC-Luc particles in the presence of AP21967 by western blot analysis (Supplementary Fig. 6B). NanoMEDIC-Luc delivered 12-fold more luciferase protein into HEK293T cells 16 hours post inoculation compared with NanoMEDIC-Luc produced in the absence of AP21967, as measured by luminescence of cell lysates (Supplementary Fig. 6C).

Next, C57BL/6 J mice were injected with a low and high dose of NanoMEDIC-Luc into the gastrocnemius muscle. Sixteen hours post injection, luciferase expression was observed in a dose-dependent manner at the injected muscle, and leakage was not detected in liver or other organs. Importantly, clearance of the luciferase protein occurred within 3 days after injection, indicating that protein delivery was transient (Fig. 6a, b).

**NanoMEDIC induces sustained exon skipping.** To assess the duration and tissue specificity of exon skipping potential of NanoMEDIC targeting human DMD sequences in vivo, a transgenic luciferase reporter mouse model was created. A single copy of the luciferase reporter gene described in Fig. 3a (promoter was switched to CAG from EF1α for body-wide constitutive expression) was inserted into Gt(ROSA)26Sor gene locus of C57BL/6 J mice (Fig. 6c). By inducing exon 45 skipping with the dual gRNA strategy we applied to DMD patient iPSCs, we investigated the exon skipping activity in vivo.

We injected NanoMEDIC containing sgRNA-DMD1 and -DMD23, respectively, into the gastrocnemius muscle of reporter mice. Luciferase expression was induced specifically at the injection site, with kinetics distinct from luciferase-loaded NanoMEDIC. A single injection of CRISPR-Cas9-loaded NanoMEDIC induced luciferase activity after 3 days, the intensity plateaued by day 7, and was sustained up to 160 days, indicating stable maintenance of genomic exon 45 skipping (Fig. 6d). Injected gastrocnemius muscle was harvested on day 189 and total RNA and genomic DNA was extracted. RT-PCR analysis confirmed 7% exon skipping efficiency in the three mice analyzed (Fig. 6e). MiSeq deep sequencing analysis revealed sharp deletion peaks at the target sites of sgRNA DMD1 and DMD23, respectively, and large deletions between the two sgRNA target sites (Supplementary Fig. 6D). Small deletions < 50 bp were predominantly observed, however, larger deletions up to 130 bp were also detected, confirming the co-delivery of both sgRNAs into the same cell (Supplementary Fig. 6E). The percentage of genomic deletion of the targeted reporter sequence was ~ 7% (Fig. 6f).

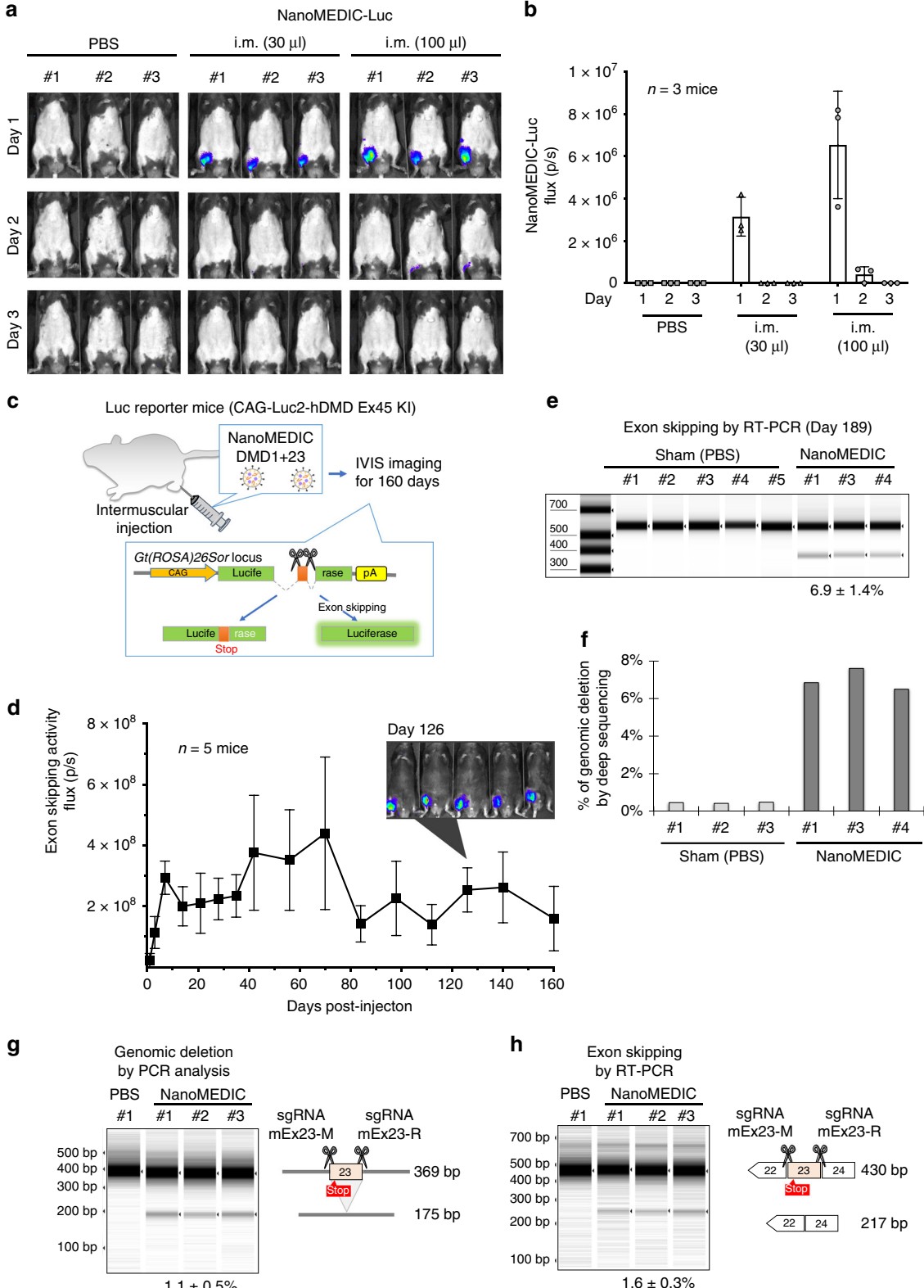

Furthermore, to target endogenous *dystrophin* gene in vivo in a mouse model of DMD, we generated two NanoMEDIC particles packaging previously reported gRNAs, targeting the *mdx* point mutation (nonsense mutation) site near the SA and the SD[6], to induce exon 23 skipping. Seven days after inoculation of the dual gRNA NanoMEDIC into tibialis anterior muscle, genomic DNA was extracted from the injected muscle tissue and analyzed by PCR. We detected 1.1% large deletion (194 bp) between the two gRNA target sites (Fig. 6g). Furthermore, the exon 23 skipping efficiency of mRNA extracted from the TA muscle of the injected mice was 1.6% (Fig. 6h).

The editing efficiency of the mouse *dystrophin* locus was lower than that of the Luc reporter, possibly owing to the difference of epigenetic status between the reporter gene locus

**Fig. 6 Transient intramuscular delivery of NanoMEDIC induces sustained genomic exon skipping in mouse models. a** Concentrated NanoMEDIC containing luciferase protein particles (NanoMEDIC-Luc) were injected into the gastrocnemius muscle of C57BL/6 J mice and visualized by IVIS imaging 1 day, 2 days, and 3 days after injection. **b** Quantification of the luciferase signal is shown in the bar graph from the three mice analyzed. Mean ± S.D. from three biological replicates. **c** Schematic depicting a transgenic mouse targeted in ROSA26 locus with a single copy of a CAG-driven luciferase coding sequence interrupted by human dystrophin exon 45, flanked by introns. Exon skipping mediated by SA or SD targeting SpCas9 RNP leads to restored luciferase expression. **d** 50 μL of RGR-DMD1 (795 ng active RNP complex) and 50 μL RGR-DMD23 (920 ng active RNP complex) NanoMEDIC were injected into the gastrocnemius muscle of the luciferase exon skipping reporter mice. The reporter luciferase signal was measured by IVIS weekly or bi-weekly from 1 to 160 days after injection and the quantified results are shown in the bar graph ($n = 5$ mice). Representative IVIS image of luciferase reporter mice 126 days after intramuscular injection with DMD1 and DMD23 NanoMEDIC is also shown. Mean ± S.D. from five biological replicates. **e** Exon skipping of the Luc reporter was verified by RT-PCR and TapeStation analysis from the gastrocnemius muscle on day 189 post injection. Mean ± S.D. from three biological replicates. **f** Percentage of genomic deletion in the Luc reporter mice was calculated by MiSeq deep sequencing analysis and CRISPResso software. **g** Two sgRNAs targeting mouse dystrophin exon 23 were packaged into NanoMEDIC (25 μL each) and injected into tibialis anterior muscle of mice, which has a nonsense mutation in exon 23. Seven days post injection, genomic DNA from the muscle was analyzed by PCR to detect the 194 bp deletion. Mean ± S.D. from three biological replicates. **h** Exon skipping of the mouse exon 23 was validated by RT-PCR and TapeStation analysis. Mean ± S.D. from three biological replicates. Source data are provided as a Source Data file.

and the endogenous gene locus. Nonetheless, our data clearly address the ability of NanoMEDIC to be functionally delivered in vivo for targeting genomic DNA and inducing exon skipping in mice.

**Development of a xeno-free NanoMEDIC production system.** To demonstrate that NanoMEDIC production is scalable and adaptable for xeno-free conditions for future clinical applications, we developed a HEK293T suspension cell production system utilizing flow electroporation by the MaxCyte STX, previously reported for large-scale lentivirus production[39]. We established a stable SV40 Large T-antigen expressing, suspension-adapted HEK293 cell line cultured in a chemically defined media. We optimized low (E4) and high (E9) electroporation energy settings in static processing assemblies for transfecting $2$–$3 \times 10^7$ HEK293T cells and treated transfected cells with benzonase (endonuclease) to remove residual plasmid DNA and enhance cell survival. We found that E9 with benzonase treatment produced higher amounts of functional NanoMEDIC particles (Supplementary Fig. 7A). NanoMEDIC produced by electroporation resulted in all-in-one particles that were dependent on Tat and AP21967 for inducing sgRNA and SpCas9 incorporation (Supplementary Fig. 7B). We then moved to large-scale flow electroporation of $> 1 \times 10^9$ HEK293T cells in 480 ml culture scale (Supplementary Fig. 7C). Produced NanoMEDIC was concentrated by overnight centrifugation and a fraction of NanoMEDIC was lysed by Triton-X to quantify the functional Cas9/sgRNA RNP complex, by utilizing recombinant SpCas9 protein and IVT sgRNA as a standard curve. As a result, we produced ~ 8.1 μg of active RNP SpCas9 complexes in total (Supplementary Fig. 7E). Although it was ~ 30% less than that produced by the same scale of adherent HEK293T culture using Lipofectamine2000 transfection (Supplementary Fig. 7E), the use of fetal bovine serum (FBS) in media should be avoided for clinical application. These results suggest that NanoMEDIC production could be scalable in suspension culture for industrial and clinical application. Moreover, when we examined the indel induction by NanoMEDIC containing 0.26 μg of active RNP complex with DMD1-sgRNA in DMD patient iPSCs, we found that it induced a higher percentage of indels versus 10 μg of electroporated recombinant RNP (Supplementary Fig. 7F). These results indicate that NanoMEDIC encapsulated CRISPR-Cas9 RNP is efficiently delivered and induces high cleavage activity in target cells.

**Discussion**
We developed an all-in-one CRISPR-Cas9 RNP delivery platform, NanoMEDIC, for genome editing in vitro and in vivo. The system relies on two homing mechanisms to package CRISPR-Cas9 protein and sgRNA separately. The first utilizes a chemical ligand-dependent incorporation of FRB-SpCas9 to FKBP12-Gag[HIV], similar to a previous report[31]. The second utilizes an HIV Ψ packaging signal to direct sgRNA flanked by HH and HDV self-cleaving ribozymes into NanoMEDIC through an interaction with Gag. In both cases, Gag is the beacon for recruitment and the dual homing approach synergistically recruits active RNP complexes into NanoMEDIC particles.

Although several CRISPR-Cas9-transient delivery systems have been reported, each system has pros and cons. For instance, to limit the expression of CRISPR SpCas9 from a lentiviral vector, the LentiSLiCES system[40] can self-inactivate because the DNA vector expressing the SpCas9 nuclease and targeting sgRNA also expresses a sgRNA targeting the viral DNA vector itself. However, the linearization of the DNA vector could lead to random integration into the genome especially at double-strand break sites.

NanoBlades has been shown to efficiently deliver RNP complexes in various cells as well as in mouse liver[29]. This system relies on the fusion of SpCas9 with Gag[MLV] and must be supplemented by wildtype Gag-Pol[MLV], meaning that there is competition between the two Gag molecules in EVs, which could reduce the number of Gag[MLV]-SpCas9 per particle. Furthermore, fusing Gag[MLV] to SpCas9 relies on MLV protease to liberate the SpCas9 nuclease from Gag, which could pose a risk of nonspecific cleavage of SpCas9 itself. When we fused SpCas9 to Gag[HIV], we found that the HIV protease cleaved cryptic peptide motifs within SpCas9 (Fig. S2C). This cleavage could be inhibited with an HIV protease inhibitor, resulting in higher activity of SpCas9 when delivered into recipient cells.

Another EV-mediated delivery system, VEsiCas9[30], relies on the stochastic incorporation of overexpressed of SpCas9 protein and sgRNA expressed in the cytoplasm of an EV producer cell by a stably expressed T7 RNA polymerase. We found that our two-mechanism loading method significantly increased packaging of all-in-one RNPs into the EV compared with stochastic incorporation of sgRNA.

The Gesicle system[31] is based on chemical-induced incorporation of the SpCas9 nuclease into secreted gesicles, except that the membrane-anchoring proteins (CherryPicker) are different. When tested with a commercially available Gesicle kit, NanoMEDIC system was more potent at inducing SSA-GFP + expression in HEK293T cells. However, we could not optimize the Gesicle production, as the kit is fixed with plasmid DNA and transfection conditions by the manufacturer. We hypothesize that one difference may also be due to inefficient incorporation of the U6-derived sgRNA in the Gesicle system. NanoMEDIC has a specific incorporation mechanism based on a packaging signal placed in

front of the gRNA, which is flanked by ribozymes and we showed that U6 promoter-driven sgRNA resulted in less-functional NanoMEDIC.

It is worth noting that NanoMEDIC utilizes HIV-1 Tat to drive the expression of sgRNA in producer cells and there is a risk of nonspecific incorporation of Tat into EVs. This poses a potential toxicity risk in recipient cells. Our proteomics data showed that although Tat protein levels were low in comparison with other host proteins incorporated into NanoMEDIC (Supplementary Table 1) and we did not observe any difference in cell proliferation of NanoMEDIC inoculated HEK293T recipient cells (Fig. 5b), we cannot completely rule out the possibility that Tat may have an effect on recipient cells.

As a proof of concept for clinical application, we targeted exon 45 in the human *dystrophin* gene to restore out-of-frame dystrophin protein by exon skipping. Simultaneous targeting of both (SA and SD) sites synergistically enhanced exon skipping activity in a Luc-based reporter system and in skeletal muscle cells differentiated from DMD patient iPSCs. NanoMEDIC delivered two sgRNAs independently without mutual inhibition, demonstrating the possibility of multiplexed genome editing. When the same combination of NanoMEDIC was injected once in luciferase reporter knock-in mice, exon skipping activity could be observed up to 160 days, indicating long-term maintenance of exon 45 skipping. Moreover, NanoMEDIC delivery in vivo was transient as luciferase protein delivered into mice was cleared within 3 days.

To expand the production scale under xeno-free conditions, we developed a suspension-based NanoMEDIC production system by flow electroporation with chemically defined media. With suspension culture, it is easier to increase the scalability of NanoMEDIC production using bioreactors. Future studies will focus on improving the purity and potency of NanoMEDIC production to target multiple muscle sites and achieve systemic delivery.

Although this study focused on packaging SpCas9/gRNA and luciferase proteins for delivery, the incorporation of other proteins of interest could be utilized. In the context of genome editing, orthologous Cas9/Cas12a nucleases or CRISPR base editors would also be candidates. As the limited genomic size restrictions of AAV would not permit CRISPR with effector molecules to be packaged unless it is split into two vectors[41], NanoMEDIC may enable the delivery of such large molecules in a protein form, which would also potentially reduce off-target risks.

## Methods

**Dimerization constructs**. FKBP12 was custom synthesized by gBlocks (Integrated DNA Technologies, Inc., Coralville, IA, USA) and cloned downstream of VSV-G to make pENTR-VSVG-FKBP12. FKBP12 was cloned upstream of a human codon optimized Gag-Pol[HIV] also containing a human Lyn mystoylation (LM) sequence to make pLM-FKBP12-Gag-pol[HIV]. Subsequently, LM-FKBP12-Gag[HIV] was PCR amplified and cloned into a pENTR vector to make pENTR-LM-FKBP12-Gag[HIV]. LM-FKBP12 from pENTR-LM-FKBP12-Gag[HIV] was PCR amplified and cloned into a pENTR vector upstream of EGFP to make pENTR-LM-FKBP12-EGFP by In-Fusion cloning. Then coding sequences in the pENTR constructs were transferred to a pHLS-EF1a-GW-A vector by LR ClonaseII reactions (ThermoFisher, Waltham, MA, USA) to generate pHLS-EF1a-VSVG-FKBP12-A, pHLS-EF1a-LM-FKBP12-Gag[HIV]-A, and pHLS-EF1a- LM-FKBP12-EGFP-A.

gBlocks FKBP12
5′-tctagaggagtgcaggtggaaaccatctccccaggagacgggcgcaccttccccaagcgcggccagacctgcg
tggtgcactacaccgggatgcttgaagatggaaagaaatttgattcctcccgggacagaaacaagcccctttaagtttatgct
aggcaagcaggaggtgatccgaggctgggaagaaggggttgcccagatgagtgtgggtcagagagccaaactgactat
atctccagattatgcctatggtgccactgggcacccaggcatcatcccaccacatgccactctcgtcttcgatgtggagctt
ctaaaactggaa-3′

FRB (T2098L) was also custom synthesized by gBlocks (Integrated DNA Technologies, Inc., Coralville, IA, USA) and cloned upstream of pDONR221-SpCas9-3 × HA (1 × NLS) (SalI) to generate pDONR221-FRB-SpCas9 or downstream of pDONR221-SpCas9 (BamHI) to generate pDONR221-SpCas9-FRB by In-Fusion Cloning (Clontech/Takara Bio USA Inc., Mountain View, CA, USA).

## Table 1 Primers used for FRB cloning.

| Primer Name | Sequence |
| --- | --- |
| FRB Mut Fragment 1F | 5′-AAAGCAGGCTGTCGAGCC-3′ |
| FRB Mut Fragment 1R | 5′-GAGGTCCTTGACATTCCCTGAT-3′ |
| FRB Mut Fragment 2 F | 5′-AAATCAGGGAATGTCAAGGACCTC GCCCAAGCCTGGGACCTCTATTATCA TGTGTTCCGACGAATCTCAAAG-3′ |
| FRB Fragment 2 R | 5′-CTTTGAGATTCGTCGGAACACATGAT AATAGAGGTCCCAGGCTTGGGCGAGG TCCTTGACATTCCCTGATTT-3′ |
| FRB Fragment 3 R | 5′-CTTATCCATGGTCGACTTTGAGATTCG TCGGAACACATG-3′ |

To generate pDONR221-FRB-SpCas9-FRB, FRB was cloned downstream of pDONR221-FRB-SpCas9 (BamHI). These coding sequences were then transferred into pHLS-EF1a-GW-A by LR Clonase II reactions (ThermoFisher Scientific, Waltham, MA, USA) to generate pHLS-EF1a-FRB-SpCas9-A, pHLS-EF1a-SpCas9-FRB-A, and pHLS-EF1a- FRB-SpCas9- FRB-A.

gBlocks FRBT2098L
5′-AAAGCAGGCTGTCGAGCCGCCACCatggcttctagaatcctctggcatgagatgtggc
atgaaggcctggaagaggcatctcgtttgtactttggggaaaggaacgtgaaaggcatgtttgaggtgctggagcccttg
catgctatgatggaacggggcccccagactctgaaggaaacatcctttaatcaggcctatggtcgagatttaatggaggcc
caagagtggtgcaggaagtacatgaaatcagggaatgtcaaggacctcctccaagcctgggacctctattatcatgtgttc
cgacgaatctcaaagTCGACCATGGATAAG-3′

To convert leucine at position 2098 to alanine (L2098A, FRB[Mut]), sense, and antisense oligos containing missense mutations to convert leucine to alanine were annealed to generate a fragment of FRB and then extended by overlap PCR to obtain full-length FRB[Mut], which was subsequently cloned into pDONR221-SpCas9-3 × HA (1 × NLS) into the SalI restriction enzyme site by In-Fusion cloning to generate pDONR221-FRB[Mut]-SpCas9. This coding sequence was then transferred into pHLS-EF1a-GW-A by an LR Clonase II reaction to generate pHLS-EF1a- FRB[Mut]-SpCas9-A (See Table 1).

**Gag[MLV]- and Gag[HIV]-fusion SpCas9 expression vectors**. Gag[MLV] was PCR amplified from pMD-MLVgag-pol[42] and cloned upstream of pHL-EF1a-SpCas9-A (SalI) by In-Fusion cloning. An MLV protease recognition site (QTSLL/TLDD) was cloned between Gag[MLV] and SpCas9. Gag[MLV]-SpCas9 was transferred to a pDONR221 vector by BP clonase reaction to generate pDONR221-Gag[MLV]-SpCas9. Then LR Clonase II reaction was performed to transfer Gag[MLV]-SpCas9 to a pHLS-EF1a-GW-A vector to generate pHLS-EF1a-Gag[MLV]-SpCas9-A.

SpCas9 was PCR amplified from pHL-EF1a-SpCas9-A and cloned into pLM-Gag-Pol[HIV] (EcoRI) between the Lyn myristoylation and matrix sequence to yield pLM-SpCas9-Gag-Pol[HIV]. An additional SV40 NLS was cloned into the 5′-end of SpCas9 and an HIV protease recognition site (SQNY/PIVQ) was cloned in between the 3′-end of SpCas9 and the 5′-end of Matrix.

**sgRNA expression constructs**. Construction of the PL-5LTR-GW-A vector was carried out by using PL-sin-EF1a-GW-iP-A as a base. PL-sin-EF1a-GW-iP was digested with MfeI and EcoRI to remove the 3′LTR, IRES-Puro, Gateway cassette, EF1α promoter, and Rev response element. Next a Gateway cassette containing the polyadenylation signal from rabbit hemoglobin beta was cloned into the vector backbone to yield PL-5LTR-GW-A. To remove the extended Psi (Ψ⁺) packaging signal from PL-5LTR-GW-A, HindIII and XbaI were used, and then the vector was religated using a 130 bp double-stranded DNA oligo nucleotide by In-Fusion cloning to generate PL-5LTR-Psi-GW-A.

**sgRNA cloning**. To clone sgRNA (targeting 5′-NNNNNNNNNNNNNNNNN NNNNnGG-3′ site) into PL-5LTR-GW-A, a spacer sequence and tracrRNA were first annealed and extension PCR was carried out with KOD Plus Neo (Toyobo) with the following conditions: 94 °C for 2 min, followed by 35 cycles of 98 °C for 10 s, 60 °C for 30 s, and 68 °C for 10 s. The first PCR product was then purified by agarose gel extraction and used as a template for overlap second PCR to add hammerhead (HH) and hepatitis delta virus (HDV) ribozymes (RGR) flanked by 15 bp overhang sequences for In-Fusion cloning into the EagI site of pENTR-AmCyan (Addgene ID: 138481). Finally, LR clonase II reaction was performed with PL-5LTR-GW-A (Addgene ID: 138480) to yield PL-5LTR-RGR(target)-AmCyan-A (i.e. Addgene ID: 138482).

First PCR for sgRNA target sequence and flanking tracrRNA scaffold sequence (See Table 2).

Second PCR for adding HH and HDR ribozyme sequences to the first PCR product. Note, 5′-XXXXXX-3′ is a reverse-complement of the first 6 bp of the target site (See Table 3).

Specific primer sequences used in this study (See Table 4).

### Table 2 Primers used for gRNA cloning in the first PCR.

| Primer Name | Sequence |
| --- | --- |
| sgRNA target-F | 5′-NNNNNNNNNNNNNNNNNNNNNGTTTTAGAGCTATGCTGGAAA-3′ |
| Sp_sgRNA | 5′-GCCCGGGTTTGAATTCAAAAAAAAAGCACCGACTCGG-3′ |
| Uni_R | 5′-TGCCACTTTTTCAAGTTGATAACGGACTAGCCTTATTTTAA-3′ |
| Primer #235 | 5′-CTTGCTATGCTGTTTCCAGCATAGCTCTAA-3′ |

### Table 3 Primers used for gRNA cloning in the second PCR.

| Primer Name | Sequence |
| --- | --- |
| sgRNA target-HH | 5′-ACCGAATTCGCGGCCXXXXXXCTGATGAGTCCGTGAGGACGAAACGAGTAAGCTCGTCNNNNNNNNNNNNNNNNNN-3′ |
| HDV Rev IF pENTR | 5′-AGTTCTAGAGCGGCCGTCCCATTCGCCATGCCGAAGCATGTTGCCCAGCCGGCGCCAGCGAGGAGGCTGGGACCATGCCGGCCAAAAAAAAAGCACCGACTCGGT-3′ |

**Construction of EGxxFP reporter vector.** Two EGFP fragments (N-terminal and C-terminal) were PCR amplified from pENTR2B-EGFP. Each fragment contains an identical 191 bp EGFP region and 15 bp overhangs with a pENTR2B vector, digested with SalI and EcoRI. The two EGFP fragments and a 70 bp PCR fragment containing the human dystrophin exon 45 sequence with an intronic region including a SA site were cloned into the digested pENTR2B vector by In-Fusion cloning to generate pENTR2B-EGxxFP-DMD-all. An LR Clonase II reaction was then carried out with this vector and pPV-EF1a-GW-iP-A to generate pPV-EF1a-EGxxFP-DMD-all-iP-A.

**Cell culture.** HEK293T cells (a kind gift from Dr. James Ellis at SickKids) were maintained in Dulbecco's Modified Eagle Medium (DMEM) high glucose (Nacalai Tesque, Kyoto, Japan) supplemented with 10% FBS (BioSera North America, Kansas City, MO, USA) and penicillin/streptomycin (Nacalai Tesque, Kyoto, Japan). Hu5 KD3 cells (a kind gift from Dr. Naohiro Hashimoto at National Center for Geriatrics and Gerontology) were maintained in DMEM, 20% FBS, 2% Ultroser G (Pall Corporation, NY, USA) and penicillin/streptomycin. U937 cells (RIKEN, Saitama, Japan) and Jurkat cells (a kind gift from Hirohide Saito, Kyoto University) were maintained in RPMI (Gibco/ThermoFisher, Waltham, MA, USA), 10% FBS, and penicillin/streptomycin. C2C12 cells were a kind gift from Hidetoshi Sakurai and maintained in DMEM, 15% FBS, 0.1 mM nonessential amino acids, 0.1 mM sodium pyruvate (Gibco/ThermoFisher, Waltham, MA, USA), 0.1 mM 2-mercaptoethanol (Gibco/ThermoFisher, Waltham, MA, USA), and penicillin/streptomycin. For differentiation, C2C12 cells were seeded at a density of 15,000 cells on a 12-well plate coated with collagen type I (Iwaki/AGC Techno Glass Co. Ltd., Shizuoka, Japan). Differentiation media (DMEM, 5% horse serum, 0.1 mM nonessential amino acids, 0.1 mM sodium pyruvate, 0.1 mM 2-mercaptoethanol, and penicillin and streptomycin) was added and the cells were differentiated for 4 days.

### Table 4 Primers used for gRNA cloning.

| Primer Name | Sequence |
| --- | --- |
| DMD#1 F[43] | 5′-GGGTATCTTACAGGAACTCCGTTTTAGAGCTATGCTGGAAA-3′ |
| sgRNA R | 5′-AAAAAAAAAGCACCGACTCGGTGCCACTTTTTCAAGTTGATAACGGACTAGCCTTATTTTAACTTGCTATGCTGTTTCCAGCATAGCTCTAA-3′ |
| DMD#23 F | 5′-AGCTGTCAGACAGAAAAAAGGTTTTAGAGCTATGCTGGAAA-3′ |
| DMD#23 RGR F | 5′-ACCGAATTCGCGGCCACAGCTCTGATGAGTCCGTGAGGACGAAACGAGTAAGCTCGTCAGCTGTCAGACAGAAAAAAG-3′ |
| SAMHD1#1 F | 5′-GTCATCGCAACGGGGACGCTGTTTTAGAGCTATGCTGGAAA-3′ |
| SAMHD1#1 RGR F | 5′-ACCGAATTCGCGGCCGATGACCTGATGAGTCCGTGAGGACGAAACGAGTAAGCTCGTCGTCATCGCAACGGGGACGCT-3′ |
| SAMHD1#2 F | 5′-CTCAAACACCCCTTCCGCAGGTTTTAGAGCTATGCTGGA-3′ |
| SAMHD1#2 HH F | 5′-ACCGAATTCGCGGCCTTTGAGCTGATGAGTCCGTGAGGACGAAACGAGTAAGCTCGTCCTCAAACACCCCTTCCGCAG-3′ |
| CCR5#1 F | 5′-GAGACCACTTGGATCC TGACATCAATTATTATACAT GTTTTAGAGCTATGCTGGAAA-3′ |
| CCR5 #1 HH F | 5′-ACCGAATTCGCGGCCATGTCACTGATGAGTCCGTGAGGACGAAACGAGTAAGCTCGTCTGACATCAATTATTATACAT-3′ |
| EGFP F | 5′-GAGACCACTTGGATCC GGGCACGGGCAGCTTGCCGG GTTTTAGAGCTATGCTGGAAA-3′ |
| EGFP HH F | 5′-ACCGAATTCGCGGCCGTGCCCCTGATGAGTCCGTGAGGACGAAACGAGTAAGCTCGTCGGGCACGGGCAGCTTGCCGG-3′ |
| VEGFA F | 5′-GAGACCACTTGGATCC GGTGAGTGAGTGTGTGCGTG GTTTTAGAGCTATGCTGGAAA-3′ |
| VEGFA HH F | 5′-ACCGAATTCGCGGCCCTCACCCTGATGAGTCCGTGAGGACGAAACGAGTAAGCTCGTCGGTGAGTGAGTGTGTGCGTG-3′ |
| EMX1F | 5′-GAGACCACTTGGATCC GAGTCCGAGCAGAAGAAGAA GTTTTAGAGCTATGCTGGAAA-3′ |
| EMX1 HH F | 5′-ACCGAATTCGCGGCCGGACTCCTGATGAGTCCGTGAGGACGAAACGAGTAAGCTCGTCGAGTCCGAGCAGAAGAAGAA-3′ |
| HDV R | 5′-AGTTCTAGAGCGGCCGTCCCATTCGCCATGCCGAAGCATGTTGCCCAGCCGGCGCCAGCGAGGAGGCTGGGACCATGCCGGCCAAAAAAAAAGCACCGACTCGGT-3′ |
| mEx23-Mdx-F | 5′-TCTTTGAAAGAGCAATAAAA GTTTTAGAGCTATGCTGGAAA-3′ |
| mEx23-R8-F | 5′-ATTTCAGGTAAGCCGAGGTTGTTTTAGAGCTATGCTGGAAA-3′ |
| Sp_sgRNA | 5′-GCCCGGGTTTGAATTCAAAAAAAAAGCACCGACTCGG-3′ |
| Uni_R | 5′-TGCCACTTTTTCAAGTTGATAACGGACTAGCCTTATTTTAA-3′ |
| Primer #235 | 5′-CTTGCTATGCTGTTTCCAGCATAGCTCTAA-3′ |
| mEx23-Mdx-HH | 5′-ACCGAATTCGCGGCCCAAAGACTGATGAGTCCGTGAGGACGAAACGAGTAAGCTCGTCTCTTTGAAAGAGCAATAAAA-3′ |
| mEx23-R8-HH | 5′-ACCGAATTCGCGGCCTGAAATCTGATGAGTCCGTGAGGACGAAACGAGTAAGCTCGTCATTTCAGGTAAGCCGAGGTT-3′ |
| HDV Rev IF pENTR | 5′-AGTTCTAGAGCGGCCGTCCCATTCGCCATGCCGAAGCATGTTGCCCAGCCGGCGCCAGCGAGGAGGCTGGGACCATGCCGGCCAAAAAAAAAGCACCGACTCGGT-3′ |

Healthy donor iPS cells (404C2 and 1383D2, and FF13096NOR) were a kind gift from Dr. Keisuke Okita, Dr. Masato Nakagawa, and Dr. Hidetoshi Sakurai at Kyoto University. DMD patient iPS cells lacking exon 44 (FFDMD111) or exon 45–46 (FF12020) were also kindly provided by Dr. Hidetoshi Sakurai. All iPSC lines were maintained in StemFit AK03N media (Ajinomoto, Tokyo, Japan) with iMatrix-511 coating (Nippi Inc., Tokyo, Japan)[19]. Establishment and use of iPSCs were made under written consent with the approval by the Ethics Committee at Faculty of Medicine or at CiRA, Kyoto University.

**Establishing EGxxFP SSA reporter cells.** piggyBac-based SSA EGFP reporter vector, pPV-EF1a-EGxxFP-DMD-all-iP-A, was transfected into C2C12, Hu5 or HEK293T cells with piggyBac transposase expressing vector pHL-EF1a-hcPBase-A[44] and then selected by puromycin for 2 weeks. The bulk reporter cells were used for NanoMEDIC inoculation experiments. For stable expression of DMD SA targeting sgRNA, PB-H1-sgRNADMD1-EF1a-RFP-IRES-Hyrogmycin was co-transfected with pHL-EF1a-hcPBase-A[44] into HEK293T EGxxFP cells and selected with hygromycin. U937 cells stably expressing SAMHD1 or EGFP cDNAs were established by electroporating U937 cells with either pPV-EF1α-SAMHD1-iP-A or pPV-EF1α-EGFP-iP-A and pHL-EF1a-hcPBase-A, and selected with puromycin.

**Differentiation of iPSCs to cortical neurons.** Neurogenin-2 (NEUROG2) inducible 404C2 iPSCs were established by electroporating 404C2 iPSCs with a doxycycline inducible NEUROG2 expressing plasmid (pPV-TetO-NEUROG2-iC-EF1a-rtTA-iP-A) and piggyBac transposase expressing vector pHL-EF1a-hcPBase-A using a NEPA21 electroporator (Nepa Gene). Cells were selected using puromycin to obtain 404C2-NEUROG2. 404C2-NEUROG2 were differentiated into cortical neurons as previously described[45]. In brief, $1.6 \times 10^5$ 404C2-NEUROG2 iPSCs were seeded into a 12-well plate coated with Matrigel (Corning Inc., Corning, NY, USA) in 1 mL of cortical neuron differentiation media (CNDM) consisting of 1:1 ratio of DMEM/F12 (Gibco/ThermoFisher, Waltham, MA, USA) and Neurobasal medium (Gibco/ThermoFisher, Waltham, MA, USA), 1% N2 supplement (Merck Millipore., Burlington, MA, USA), 2% B27 supplement (Gibco/ThermoFisher, Waltham, MA, USA), 10 ng/mL brain-derived neurotrophic factor (R&D Systems, Minneapolis, MN, USA), 10 ng/mL glial cell-derived neurotropic factor (R&D Systems, Minneapolis, MN, USA), 10 ng/mL Neurotrophin-3 (NT-3) and 1 μg/mL doxycycline. Half CDM media changes were carried out every day for seven days.

**Differentiation of iPSCs to skeletal muscle cells.** Dox-inducible MYOD1 expressing iPSCs were established and differentiated to skeletal muscle cells as previously described[46]. In brief on Day 1, $3 \times 10^5$ iPSCs were seeded onto a six-well plate coated with Matrigel in AK03N StemFit media. On Day 2, media was changed to Primate ES Cell Media (ReproCELL Inc., Kanagawa, Japan). On Day 3, media was changed to Primate ES Cell Media with 1 μg/mL doxycycline. On Day 4, media was changed to skeletal muscle cell differentiation media (SMCDM) consisting of 5% KSR in Alpha-mEM, 200 μM 2-mercaptoethanol, and 1 μg/mL doxycycline. Media was changed every day to the SMCDM until Day 6–7.

**NanoMEDIC production from adherent cell culture.** HEK293T cells were seeded in 10 cm plates at a density of $2–3 \times 10^6$ cells per plate. The next day the cells were transfected with 10 μg of pHLS-EF1a-FKBP12-Gag[HIV](Addgene ID: 138476), 10 μg pHLS-EF1a-FRB-SpCas9-A (Addgene ID: 138477), 10 μg PL-5LTR-RGR(Target)-AmCyan-A (i.e. Addgene ID: 138482), 2 μg of pcDNA1-Tat[HIV] (Addgene ID: 138478), and 5 μg pVSV-G (Addgene ID: 138479) by Lipofectamine 2000 (ThermoFisher Scientific, Waltham, MA, USA). The following day, the transfection media was replaced with 10 mL of fresh DMEM media containing 10 % FCS and 300 nM of AP21967 (Clontech/Takara Bio USA Inc., Mountain View, CA, USA). Supernatant was harvested 36–48 hours after transfection, filtered through a 0.45 μm syringe filter (Sartorius, Göttingen, Germany), and then centrifuged at $8,800 \times g$'s overnight (4 °C) in an Avanti JXN-30 centrifuge and JS-24.38 rotor (Beckman Coulter, Brea, CA, USA). The pelleted EVs were resuspended in 100 μL of chilled HBSS or phosphate-buffered saline (PBS; ThermoFisher Scientific, Waltham, MA, USA) and then aliquoted into 1.5 mL centrifuged tubes and stored at −80 °C. For in vivo inoculation experiments targeting the Luc exon skipping reporter and mouse Dystrophin exon 23 in mice, 72 mL of NanoMEDIC were concentrated by two rounds of centrifugations ($8,800 \times g$, 13–14 hours, 4 °C) and resuspended in 150 μL of chilled PBS.

**NanoMEDIC production from a suspension system.** HEK293 Gibco Viral Production Cells (Cat# A35684, ThermoFisher Scientific, Waltham, MA, USA) were stably transduced with a pPV-EF1a-SV40TAg-iP-A piggyBac vector by puromycin to obtain SV40 large T-antigen stable expressing cells (HEK293T-GVPC). These cells were then expanded in Nalgene Single-Use PETG Erlenmeyer Flasks in LV Max Production Media (Gibco/ThermoFisher Scientific, Waltham, MA, USA). Before electroporation, HEK293T-GVPC were centrifuged at $200 \times g$'s for 10 minutes, washed once in MaxCyte electroporation buffer, and centrifuged again at $200 \times g$'s for 10 minutes. The buffer was aspirated and then the cells were resuspended at $8 \times 10^8$ cells/mL.

For small-scale static electroporation with a MaxCyte STX, OC-400 processing assemblies were used to electroporate $3 \times 10^7$ cells with 32.4 μg of pHLS-EF1a-FKBP12-Gag[HIV], 32.4 μg pHLS-EF1a-FRB-SpCas9-A, 32.4 μg PL-5LTR-RGR-AmCyan-A, 6.5 μg of pcDNA1-Tat[HIV], and 3.6 μg pVSV-G at optimization energy 9. After electroporation, the cells were incubated with benzonase (Merck, Kenilworth, NJ, USA) in a final concentration of 5 mM $MgCl_2$ in a 37 °C humidified incubator with 5% $CO_2$ for 40 mins. Finally, 20 mL of LV Max Production Media containing 300 nM of AP21967 was added to the cells and then cultured in a 125 mL Nalgene Single-Use PETG Erlenmeyer Flask in a 37 °C humidified on an orbital shaker at 125 RPM. Supernatant was harvested 36–48 hours after transfection, filtered through a 0.45 μm syringe filter, and then centrifuged at $8,800 \times g$'s overnight in an Avanti JXN-30 centrifuge (Beckman Coulter, Brea, CA, USA). The pelleted EVs were resuspended in 100 μL of HBSS or PBS (Gibco/ThermoFisher Scientific, Waltham, MA, USA) and then aliquoted into 1.5 mL microcentrifuge tubes and stored at –80 °C.

For large-scale flow electroporation with a MaxCyte STX, a CL-2 processing assembly was used to electroporate ~ $1.2 \times 10^9$ cells with 1038 μg of pHLS-EF1a-FKBP12-Gag[HIV], 1038 μg pHLS-EF1a-FRB-SpCas9-A, 1038 μg PL-5LTR-RGR-AmCyan-A, 208 μg of pcDNA1-Tat[HIV], and 51 μg pVSV-G at optimization energy 9. After electroporation, the cells were incubated in with benzonase in a final concentration of 5 mM $MgCl_2$ in a 37 °C humidified incubator with 5% $CO_2$ for 40 mins. Finally, 480 mL of LV Max Production Media containing 300 nM of AP21967 was added to the cells and then split into two 1 L single-use PETG Erlenmeyer Flasks and cultured in a 37 °C humidified on an orbital shaker at 100 RPM.

**NanoMEDIC inoculation.** For HEK293T, C2C12, and Hu5 EGxxFP reporter cells, $2.5 \times 10^4$ or $5.0 \times 10^4$ cells were seeded in a 48-well plate. The following day, the reporter cells were inoculated with ~ 10 μL of 80-fold concentrated NanoMEDIC particles, unless indicated otherwise, and then analyzed by flow cytometry 3 days post inoculation using an BD LSRFortessa Flow Cytometer. For iPSC inoculation, $1 \times 10^5$ cells were inoculated with NanoMEDIC in a 24-well plate.

**Determination of mutation frequency.** Genomic DNA was extracted from cells by a MonoFas Genomic DNA Extraction kit (GL Sciences Inc., Tokyo, Japan) according to the manufacturer's protocol. In total, 100 ng of genomic DNA was then PCR amplified using a PrimeSTAR GXL DNA Polymerase (Takara Bio USA Inc., Mountain View, CA, USA) and appropriate primers (Supplemental Table 1). Thermocycle conditions were as follows: 94 °C for 2 min, followed by 35 cycles of 98 °C for 10 sec, 60 °C for 15 sec and 68 °C for 30 sec. The resulting PCR product was column purified using a Wizard SV Gel and PCR Clean-up System (Promega, Madison, WI, USA). Then 400 ng of PCR product was denatured and reannealed using a thermocycler to heat the samples to 95 °C and then gradually cool to 4 °C in a volume of 19 μL containing 1 × Buffer 2.1 (New England Biolabs., Ipswich, MA, USA). Next, 1 μL of T7 Endonuclease I (T7E1; NEB, 10 units/μL) enzyme was added to initiate the reaction at 37 °C for 15 minutes. EDTA (Nacalai Tesque, Kyoto, Japan) was added to stop the reaction at a final concentration of 6 mM. The samples were then analyzed for cleavage products using D1000 High Sensitivity ScreenTapes on a 2200 TapeStation (Agilent Technologies, Santa Clara, CA, USA). The mutation frequency, % indels, was determined as previously reported using the following equation %indels = 100 × $(1 –(1 – f)^{1/2})$, where $f$ represents the fraction of cleaved PCR product. The primer sequences used for PCR amplification are listed in Table 5.

### Table 5 Primers used for T7EI assay.

| Primer Name | Sequence |
|---|---|
| DMD1 specific amplicon | Fwd: 5′-CCCTGACACATAAAAGGTGTCTTTCTGT-3′ |
| | Rev: 5′-TTCTGTCTGACAGCTGTTTGCAGAC-3′ |
| DMD23 specific amplicon | Fwd: 5′-AAAAATTGGGAAGCCTGAATCTGC-3′ |
| | Rev: 5′-AAGAAAGCTTAAAAAGTCTGCTAAAATGTTTT-3′ |
| DMD amplicon for looking at exon 45 removal | Fwd: 5′-TACAACTGCATGTGGTAGCACACTG-3′ |
| | Rev: 5′-CATTCCTATTAGATCTGTCGCCCTAC-3′ |
| SAMHD1 #1 and #2 amplicon | Fwd: 5′-CGCCGAGGTTCTTGACTGC-3′ |
| | Rev: 5′-CTCGGATGTTCTTCAGCAGCA-3′ |
| CCR5 amplicon | Fwd: 5′-ATGTATAAAACAGTTTGCATTCATGGAGGG-3′ |
| | Rev: 5′-CATGATGGTGAAGATAAGCCTCACAG-3′ |
| EMX1 amplicon | Fwd: 5′-CTGCCATCCCCTTCTGTGAATGT-3′ |
| | Rev: 5′-GGAATCTACCACCCCAGGCTCT-3′ |
| VEGFA amplicon | Fwd: 5′-GCATACGTGGGCTCCAACAGGT-3′ |
| | Rev: 5′-CCGCAATGAAGGGGAAGCTCGA-3′ |

For TIDE analysis, Sanger sequencing of the amplified genomic DNA region of interest was carried out and then the resulting trace files were used to calculate the % indels by the TIDE webtool.

**Proteome analysis.** NanoMEDIC particles or exosomes were collected from transfected HEK293T cells by overnight centrifugation at $8,800 \times g$'s at 4 °C. The media was discarded, the pellet was washed with PBS (Nacalai Tesque, Kyoto, Japan), and then the pellet was recentrifuged at $100,000 \times g$'s for 3 hours at 4 °C. The PBS was decanted and resuspended in 100 µL of PBS. This solution was directly lysed with ice cold PTS buffer (12 mM Sodium deoxycholate, 12 mM sodium lauroyl sarcosinate, 100 mM Tris-HCl (pH 9.0)). These protein samples were subjected to reduction, alkylation, Lys-C/trypsin digestion (enzyme ratio: 1/100) and desalting using StageTip[47]. Resulting peptides were labeled with isobaric tags for relative and absolute quantification (iTRAQ, Sciex, Framingham, MA, USA) and mixed in loading buffer (0.5% trifluoroacetic acid and 4% (v/v) acetonitrile). They were subsequently subjected to nanoLC-MS/MS using a TripleTOF 5600 System (AB Sciex) equipped with an HTC-PAL autosampler (CTC Analytics). Loaded peptides were separated on a self-pulled analytical column (150-mm length, 100-µm i.d.) using a Dionex UltiMate 3000 RSLCnano System. The mobile phases were composed of 0.5% acetic acid with 5% (v/v) DMSO (solution A) and 0.5% acetic acid in 80% (v/v) acetonitrile with 5% (v/v) DMSO (solution B)[48]. A two-step gradient condition of 10–40% (v/v) solution B for 120 min and 40–100% (v/v) solution B for 5 min was used with a flow rate of 400 µL/min. The applied spray voltage was 2300 V, and the MS scan range was $300-1500$ m/z every 0.25 s.

The raw data files were analyzed using ProteinPilot v5.0 (Sciex, Framingham, MA, USA) with acceptable modifications of N-terminal iTRAQ, iTRAQ of lysine, carbamidomethylation of cysteine, oxidation of methionine, phosphorylation of serine, threonine or tyrosine, deamidation of asparagine or glutamine, the N-terminal pyro-glutamic acid of glutamine or glutamic acid, and protein N-terminal acetylation. Peak lists, which were generated from a ProteinPilot.group file, were analyzed by Mascot v2.5 (Matrix Science Inc., Boston, MA, USA) with the carbamidomethylation of cysteine as the fixed modification, and the N-terminal iTRAQ, iTRAQ of lysine, and methionine oxidation as the variable modification. Both database search engines were used against human entries of UniProt/Swiss-Prot release 2016_06 (8-June-2016) with cargo proteins incorporated from the plasmid sequences. We allowed a precursor mass tolerance of 20 ppm, a fragment ion mass tolerance of 0.1 Da, and used strict trypsin and Lys-C specificity, which allowed up to two missed cleavages. For the peptide identification, peptides were rejected if any of the following conditions were not satisfied: (a) if the same scan was assigned to different peptides between ProteinPilot and Mascot, (b) peptide confidence was below 0.05, (c) the charge state was more than 5, (d) or the peptide length was less than six amino acids. For the protein identification, at least two confidently ($p < 0.05$) identified peptides per protein were used. Single peptides with higher confidence ($p < 0.01$) were allowed. Finally, peptides were grouped into protein groups based on previously established rules[49]. False discovery rates were estimated by searching against a decoy sequence database ($< 1\%$). For the peptide and protein quantification, we used RiMS approach to increase the accuracy[50].

The iTRAQ area of proteins in exosomes was plotted against those in NanoMEDIC particles in a scatterplot. In addition, proteins identified in NanoMEDIC particles were analyzed in DAVID Bioinformatics Resources v6.8[51]. The MS/MS data have been deposited to the ProteomeXchange Consortium via jPOSTrepo (https://repository.jpostdb.org/) with the dataset identifier JPST000623 (PXD014527).

**Determination of human DMD exon skipping efficiency.** RNA was extracted from skeletal muscle cells differentiated from DMD patient iPSCs using a NucleoSpin RNA kit (Macherey-Nagel GmbH & Co. KG., Düren, Germany) according to the manufacturer's instructions. Approximately 500 ng of RNA was reverse transcribed into cDNA using a ReverTra Ace RT Mix (Toyobo, Osaka, Japan) according to the manufacturer's instructions. Then 50 ng of cDNA was PCR amplified by PrimeStar GXL and the PCR product was cleaned by a Wizard SV Gel and PCR Clean-Up System (Promega, Madison, WI, USA). Following the PCR, 4 ng of PCR product was analyzed using D1000 High Sensitivity ScreenTapes on a 2200 TapeStation. The area resulting lower band was quantified as a fraction of the total area from the upper and lower band added together to estimate the exon skipping frequency.

**Protein analysis.** For western blot analysis of EVs, an equal volume of EVs and 2 × sample buffer (Bio-Rad), containing 50 mM DTT (Nacalai Tesque, Kyoto, Japan) were mixed, and heated at 95 °C for 5 minutes. Next, samples were incubated on ice for 2 minutes and then loaded onto a precast Bis-Tris 8–16% polyacrylamide gel (GeneScript, Piscataway, NJ, USA) and run for 90 min at 120 V. The gel was pre-equilibrated in blotting buffer before being transferred to a 0.2 µm nitrocellulose membrane using an iBlot dry blotting system (Invitrogen/ThermoFisher Scientific, Waltham, MA, USA). The membrane was blocked for 30 minutes at room temperature in Blocking One solution (Nacalai Tesque, Kyoto, Japan). Then the membrane was incubated at 4 °C overnight in Can Get Signal Solution 1 (Toyobo, Osaka, Japan) with an appropriate primary antibody

as described below. The following day, the membrane was washed five times for 5 minutes each in TBS containing 0.5% Tween-20 (Nacalai Tesque, Kyoto, Japan), and incubated with either anti mouse IgG or anti rabbit IgG, HRP-linked secondary antibody (Cat# 7076 or 7074, Cell Signaling Technology, Danvers, MA, USA). The membrane was washed five times for 5 minutes tris-buffered saline (TBS) containing 0.5% Tween-20. Finally, the membrane was incubated in ECL Prime Western Blotting and imaged on an ImageQuant LAS 4000 or a Bio-Rad ChemiDoc XRS+ System. Membranes were probed with 100,000-fold diluted monoclonal anti-VSV-G (Cat# V5507, Sigma-Aldrich/Merck, St. Louis, MO, USA), 1000-fold diluted monoclonal anti-HIV p24 (Cat# ab9071, Abcam, Cambridge, UK), 1000-fold diluted monoclonal anti-FKBP12 (Cat# 635089, Clontech/Takara Bio USA Inc., Mountain View, CA, USA), 1,000-fold diluted monoclonal anti-FRB (Cat# 635091, Clontech/Takara Bio USA Inc., Mountain View, CA, USA), 1000-fold diluted anti-SpCas9 (Cat#61577, Active Motif), or 1000-fold diluted anti-HA (Cat# sc-7392, Santa Cruz Biotechnologies) antibody.

For dystrophin and myosin heavy chain protein analyses, a 66–440 kDa Wes Separation module (Protein Simple, San Jose, California, USA) was used. Samples were prepared according to the manufacturer's instructions.

**In vitro Cas9 DNA cleavage activity assay.** Concentrated EVs were lysed in 2 × lysis buffer (40 mM HEPES (Nacalai Tesque, Kyoto, Japan), pH 7.5, 200 mM KCl (Nacalai Tesque, Kyoto, Japan), 10 mM MgCl₂ (Nacalai Tesque, Kyoto, Japan), 2 mM DTT, 10% glycerol, 0.2% Triton X-100) for 1 hour on ice. Then 9 µL of lysate was added to 3 µL of 5 × cleavage buffer (100 mM HEPES, pH 7.5, 750 mM KCl, 5 mM DTT, 5 mM MgCl₂, 250 µg/mL BSA (Sigma-Aldrich/Merck, St. Louis, MO, USA), and 50 % glycerol), 1 µL of DNA template (10 ng/µL), and 1 µL of water. The tube was incubated at 37 °C for one hour. Next, 0.5 µL of RNase A (Invitrogen/ThermoFisher Scientific, Waltham, MA, USA) was added and incubated at 37 °C for 30 min. Then, 0.5 µL of proteinase K (Nacalai Tesque, Kyoto, Japan) was added to the reaction mixture and incubated at 50 °C for 20 min. Finally, 2 µL of the reaction mixture was diluted with 2 µL of High Sensitivity Sample Buffer and loaded onto a 2200 TapeStation (Agilent Technologies, Santa Clara, CA, USA). Active RNP complexes from NanoMEDIC was quantified by comparing Nano-MEDIC cleavage activity with a standard RNP curve consisting of WT SpCas9 recombinant protein (PNA Bio Inc., Thousand Oaks, CA, USA or Integrated DNA Technologies, Inc., Coralville, IA, USA) and in vitro transcribed sgRNA (Megashortscript Kit, Thermo Fisher Scientific, Waltham, MA, USA) targeting the same cleavage site.

**In vivo luciferase delivery experiments.** For NanoMEDIC-Luc delivery experiment, NanoMEDIC-Luc (30 or 100 µL) were injected into gastrocnemius muscles of C57BL/6 J mice and luciferase luminescence signal was visualized by IVIS imaging 1 day, 2 days, and 3 days after injection. CAG-Luc2-hDMD Ex45 KI mice were established by targeting the Gt(ROSA)26Sor locus with pCAGGS-Luc2-hEx45 reporter construct using CRISPR-Cas9 (performed by Axcelead Drug Discovery Partners Inc., Kanagawa, Japan). For luciferase imaging experiments, mice were anesthetized with isoflurane and injected 3 mg of Luciferin (diluted in PBS, Caliper/PerkinElmer, Waltham, MA, USA) 5–15 minutes prior to the analysis of Luciferase expression. Then the mice were transferred to an imaging chamber in an IVIS Spectrum In Vivo Imaging System (PerkinElmer, Waltham, MA, USA). All in vivo experiments were approved by the Institutional Animal Care and Use Committee in Takeda Pharmaceutical Company Limited (Approval number AU-00020951).

**Evaluation of in vivo exon skipping efficiencies.** NanoMEDIC containing SpCas9 protein and Ex45 gRNAs (DMD1 and DMD23) were intramuscularly administered to CAG-Luc2-hDMD Ex45 KI reporter mice at the dosage of 795 ng (DMD1) or 920 ng (DMD23) in 100 µL, respectively. Gastrocnemius (GC) muscles were collected 189 days after the single administration. RNAs were extracted from GC muscles using RNeasy Fibrous Tissue Kit (Qiagen, Hilden, Germany) and reverse transcribed using High Capacity RNA-to-cDNA Kit (Thermo Fisher Scientific, Waltham, MA, USA). These cDNAs were amplified by PCR primers (5′-TGCCCACACTATTTAGCTTC-3′ and 5′-GTCGATGAGAGCGTTTGTAG-3′) using Q5 HotStart High-Fidelity 2 × Master Mix (New England Biolabs, Ipswich, MA, USA). PCR products were purified by QIAquick PCR Purification Kit (Qiagen, Hilden, Germany) and subjected to electrophoresis using 4200 TapeStation (Agilent Technologies, Santa Clara, CA, USA). Exon skipping efficiencies were determined by calculating the ratio of unskipped (553 bp) and skipped (377 bp) PCR products.

**Evaluation of in vivo genomic indel patterns by deep sequencing.** Genomic DNAs were extracted and the human DMD exon 45 sequence in the Luc reporter region was PCR amplified by the 1st primer pair (MiSeq-DMD-Rd1-fwdX and MiSeq-DMD-Rd2-revX) with unique 4 bp index sequence per sample (see the table below). Then, the 1st PCR products were PCR amplified by the 2nd primer pair (MiSeq-P5-N50X-fwd and MiSeq-P7-N70XR-rev) using polymerase. The resultant PCR amplicons were gel extracted and quantified by the KAPA Library Quantification Kit for Illumina (KAPA Biosystems, Wilmington, MA, USA). The samples

**Table 6 Primers used for deep sequencing analysis.**

| Primer name | Sequence |
|---|---|
| 1st PCR primers | |
| MiSeq-DMD-Rd1-fwd1-AGTC | 5′-CTCTTTCCCTACACGACGCTCTTCCGATCTagtcAATAAAAAGACATGGGGCTTCA-3′ |
| MiSeq-DMD-Rd1-fwd2-GTCA | 5′-CTCTTTCCCTACACGACGCTCTTCCGATCTgtcaAATAAAAAGACATGGGGCTTCA-3′ |
| MiSeq-DMD-Rd1-fwd3-TCAG | 5′-CTCTTTCCCTACACGACGCTCTTCCGATCTtcagAATAAAAAGACATGGGGCTTCA-3′ |
| MiSeq-DMD-Rd1-fwd4-CAGT | 5′-CTCTTTCCCTACACGACGCTCTTCCGATCTcagtAATAAAAAGACATGGGGCTTCA-3′ |
| MiSeq-DMD-Rd1-fwd5-ACTG | 5′-CTCTTTCCCTACACGACGCTCTTCCGATCTactgAATAAAAAGACATGGGGCTTCA-3′ |
| MiSeq-DMD-Rd1-fwd6-CTGA | 5′-CTCTTTCCCTACACGACGCTCTTCCGATCTctgaAATAAAAAGACATGGGGCTTCA-3′ |
| MiSeq-DMD-Rd1-fwd7-TGAC | 5′-CTCTTTCCCTACACGACGCTCTTCCGATCTtgacAATAAAAAGACATGGGGCTTCA-3′ |
| MiSeq-DMD-Rd1-fwd8-GACT | 5′-CTCTTTCCCTACACGACGCTCTTCCGATCTgactAATAAAAAGACATGGGGCTTCA-3′ |
| MiSeq-DMD-Rd1-fwd9-ATCG | 5′-CTCTTTCCCTACACGACGCTCTTCCGATCTatcgAATAAAAAGACATGGGGCTTCA-3′ |
| MiSeq-DMD-Rd1-fwd10-TCGA | 5′-CTCTTTCCCTACACGACGCTCTTCCGATCTtcgaAATAAAAAGACATGGGGCTTCA-3′ |
| MiSeq-DMD-Rd1-fwd11-CGAT | 5′-CTCTTTCCCTACACGACGCTCTTCCGATCTcgatAATAAAAAGACATGGGGCTTCA-3′ |
| MiSeq-DMD-Rd1-fwd12-GATC | 5′-CTCTTTCCCTACACGACGCTCTTCCGATCTgatcAATAAAAAGACATGGGGCTTCA-3′ |
| MiSeq-DMD-Rd1-fwd13-ATGC | 5′-CTCTTTCCCTACACGACGCTCTTCCGATCTatgcAATAAAAAGACATGGGGCTTCA-3′ |
| MiSeq-DMD-Rd1-fwd14-TGCA | 5′-CTCTTTCCCTACACGACGCTCTTCCGATCTtgcaAATAAAAAGACATGGGGCTTCA-3′ |
| MiSeq-DMD-Rd1-fwd15-GCAT | 5′-CTCTTTCCCTACACGACGCTCTTCCGATCTgcatAATAAAAAGACATGGGGCTTCA-3′ |
| MiSeq-DMD-Rd1-fwd16-CATG | 5′-CTCTTTCCCTACACGACGCTCTTCCGATCTcatgAATAAAAAGACATGGGGCTTCA-3′ |
| MiSeq-DMD-Rd1-fwd17-AGCT | 5′-CTCTTTCCCTACACGACGCTCTTCCGATCTagctAATAAAAAGACATGGGGCTTCA-3′ |
| MiSeq-DMD-Rd1-fwd18-GCTA | 5′-CTCTTTCCCTACACGACGCTCTTCCGATCTgctaAATAAAAAGACATGGGGCTTCA-3′ |
| MiSeq-DMD-Rd2-rev1-AGTC | 5′-CTGGAGTTCAGACGTGTGCTCTTCCGATCTagtcCCTTTCACCCTGCTTATAATCTC-3′ |
| MiSeq-DMD-Rd2-rev2-GTCA | 5′-CTGGAGTTCAGACGTGTGCTCTTCCGATCTgtcaCCTTTCACCCTGCTTATAATCTC-3′ |
| MiSeq-DMD-Rd2-rev3-TCAG | 5′-CTGGAGTTCAGACGTGTGCTCTTCCGATCTtcagCCTTTCACCCTGCTTATAATCTC-3′ |
| MiSeq-DMD-Rd2-rev4-CAGT | 5′-CTGGAGTTCAGACGTGTGCTCTTCCGATCTcagtCCTTTCACCCTGCTTATAATCTC-3′ |
| MiSeq-DMD-Rd2-rev5-ACTG | 5′-CTGGAGTTCAGACGTGTGCTCTTCCGATCTactgCCTTTCACCCTGCTTATAATCTC-3′ |
| MiSeq-DMD-Rd2-rev6-CTGA | 5′-CTGGAGTTCAGACGTGTGCTCTTCCGATCTctgaCCTTTCACCCTGCTTATAATCTC-3′ |
| MiSeq-DMD-Rd2-rev7-TGAC | 5′-CTGGAGTTCAGACGTGTGCTCTTCCGATCTtgacCCTTTCACCCTGCTTATAATCTC-3′ |
| MiSeq-DMD-Rd2-rev8-GACT | 5′-CTGGAGTTCAGACGTGTGCTCTTCCGATCTgactCCTTTCACCCTGCTTATAATCTC-3′ |
| MiSeq-DMD-Rd2-rev9-ATCG | 5′-CTGGAGTTCAGACGTGTGCTCTTCCGATCTatcgCCTTTCACCCTGCTTATAATCTC-3′ |
| MiSeq-DMD-Rd2-rev10-TCGA | 5′-CTGGAGTTCAGACGTGTGCTCTTCCGATCTtcgaCCTTTCACCCTGCTTATAATCTC-3′ |
| MiSeq-DMD-Rd2-rev11-CGAT | 5′-CTGGAGTTCAGACGTGTGCTCTTCCGATCTcgatCCTTTCACCCTGCTTATAATCTC-3′ |
| MiSeq-DMD-Rd2-rev12-GATC | 5′-CTGGAGTTCAGACGTGTGCTCTTCCGATCTgatcCCTTTCACCCTGCTTATAATCTC-3′ |
| MiSeq-DMD-Rd2-rev13-ATGC | 5′-CTGGAGTTCAGACGTGTGCTCTTCCGATCTatgcCCTTTCACCCTGCTTATAATCTC-3′ |
| MiSeq-DMD-Rd2-rev14-TGCA | 5′-CTGGAGTTCAGACGTGTGCTCTTCCGATCTtgcaCCTTTCACCCTGCTTATAATCTC-3′ |
| MiSeq-DMD-Rd2-rev15-GCAT | 5′-CTGGAGTTCAGACGTGTGCTCTTCCGATCTgcatCCTTTCACCCTGCTTATAATCTC-3′ |
| MiSeq-DMD-Rd2-rev16-CATG | 5′-CTGGAGTTCAGACGTGTGCTCTTCCGATCTcatgCCTTTCACCCTGCTTATAATCTC-3′ |
| MiSeq-DMD-Rd2-rev17-AGCT | 5′-CTGGAGTTCAGACGTGTGCTCTTCCGATCTagctCCTTTCACCCTGCTTATAATCTC-3′ |
| MiSeq-DMD-Rd2-rev18-GCTA | 5′-CTGGAGTTCAGACGTGTGCTCTTCCGATCTgctaCCTTTCACCCTGCTTATAATCTC-3′ |
| 2nd PCR primers | |
| MiSeq-P5-N501-fwd | 5′-AATGATACGGCGACCACCGAGATCTACAtagatcgcCTCTTTCCCTACACGACGCTC-3′ |
| MiSeq-P5-N502-fwd | 5′-AATGATACGGCGACCACCGAGATCTACActctctatCTCTTTCCCTACACGACGCTC-3′ |
| MiSeq-P5-N503-fwd | 5′-AATGATACGGCGACCACCGAGATCTACAtatcctctCTCTTTCCCTACACGACGCTC-3′ |
| MiSeq-P7-N701R-rev | 5′-CAAGCAGAAGACGGCATACGAGATGTGAtcgccttaCTGGAGTTCAGACGTGTGCTC-3′ |
| MiSeq-P7-N702R-rev | 5′-CAAGCAGAAGACGGCATACGAGATGTGActagtacgCTGGAGTTCAGACGTGTGCTC-3′ |
| MiSeq-P7-N703R-rev | 5′-CAAGCAGAAGACGGCATACGAGATGTGAttctgcctCTGGAGTTCAGACGTGTGCTC-3′ |
| MiSeq-P7-N704R-rev | 5′-CAAGCAGAAGACGGCATACGAGATGTGAgctcaggaCTGGAGTTCAGACGTGTGCTC-3′ |
| MiSeq-P7-N705R-rev | 5′-CAAGCAGAAGACGGCATACGAGATGTGAaggagtccCTGGAGTTCAGACGTGTGCTC-3′ |
| MiSeq-P7-N706R-rev | 5′-CAAGCAGAAGACGGCATACGAGATGTGAcatgcctaCTGGAGTTCAGACGTGTGCTC-3′ |

were diluted to 2 nm and treated with 0.2 N NaOH for 5 min. Denatured samples were mixed equally to a final concentration of 12 pm, added 3 pm of PhiX control, and run on MiSeq with the MiSeq Reagent Kit v3 for 600 cycles (300 bp × 2). From the resultant FASTQ sequence data, illumine adapter sequences and low-quality reads (BQ < 20) were removed by *cutadapt* software (https://cutadapt.readthedocs.io/), samples were split by *fastx_barcode_splitter* software (http://hannonlab.cshl.edu/fastx_toolkit/), pair-mate reads were extracted by *fastq_pair* (https://github.com/linsalrob/fastq-pair), overlapped forward and reverse reads were merged by *flash2* (https://github.com/dstreett/FLASH2), 4 bp index and primer sequences were removed by *cutadapt*. Finally, the sequencing reads were mapped to human DMD sequence and indel patterns were analyzed by *CRISPResso* software[52]. The data sets generated during the current study are available in the NCBI SRA repository [PRJNA560477] (See Table 6).

**In vivo mouse dystrophin exon 23 skipping experiments**. For mouse *dystrophin* exon 23 skipping experiments, NanoMEDIC containing SpCas9 protein and gRNAs (mExon 23-M and mExon 23-R) were injected into tibialis anterior muscle of NOG-mdx (NOD.Cg-Prkdc^scid Il2rg^tm1Sug Dmd^mdx /Jic) (Ito et al. Blood, 2002, Bulfield et al. PNAS., 1984) mouse, which has a nonsense mutation in exon 23 of *dystrophin* gene. The dosage of Cas9 protein amount in NanoMEDIC was 532 ng for mExon 23-M (target site: TCTTTGAAAGAGCAAtAAAATGG) and 1010 ng for mExon 23-R (target site: ATTTCAGGTAAGCCGAGGTTTGG) in total 50 µL injection volume, respectively. Seven days after the injection, mice were sacrificed and the injected muscle tissues were corrected. Genomic cleavage was assessed by PCR using the following primers (5′-GAAACTCATCAAATATGCGTGTTAGTG-3′ and 5′- AATATCTTTGAAGGACTCTGGGTAAA-3′), and exon skipping activity was measured by RT-PCR using the following primers (5′-GGATCCAGCAGTCAGAAAGC-3′ and 5′-TCACCAACTAAAAGTCTGCATTG-3′). The mouse dystrophin experiments were approved by the CiRA Animal Experiment Committee in Kyoto University (Approval number KEI19-125).

**NanoMEDIC purification**. NanoMEDIC was harvested as described above and incubated with benzonase for 30 minutes at 37 °C to remove residual DNA plasmids. A CIMmultus QA-8 mL (BIA Separations, Ajdovščina, Slovenia) column was equilibrated with 80 mL of Buffer A (20 mm Tris-HCl, pH 8.0, and 0.1 M NaCl) and the sample was loaded onto the column and then washed with 135 mL of Buffer A, collected in three 45 mL fractions. Next 170 mL of Buffer B (20 mm Tris-HCl, pH 8.0, and 0.2 M NaCl) was run through the column and collected in four 45 mL fractions. For elution of NanoMEDIC, 200 mL of Buffer C (20 mm Tris-HCl, pH 8.0, and 0.65 M NaCl) was run through the column and then collected in two

100 mL fractions. Finally, 45 mL of Buffer D (20 mᴍ Tris-HCl, pH 8.0, and 1 ᴍ NaCl) was run through the column and collected in one tube. The elutions were checked by western blot for SpCas9 and p24 to determine which fractions NanoMEDIC was eluted in and then elutions were concentrated by overnight centrifugation as described above and resuspended in PBS.

**Electron microscopy analysis.** The purified NanoMEDIC particles were fixed in 1% paraformaldehyde, negatively stained with 2% phosphotungstic acid, and observed by transmission electron microscopy (HT-7700, Hitachi, Tokyo, Japan) operating at 80 kV with XR81-B CCD camera. For particle size measurement, "fit ellipse" function of ImageJ software[53] was used to regard the particle shape as ellipse. Then, the size distribution was measured for about a hundred particles by using ImageJ software.

**NanoSight nanoparticle tracking analysis.** NanoMEDIC samples were diluted by 10,000-fold and applied to a NanoSight LM10 (Malvern Panalytical Ltd., Malvern, UK) chamber. Five recordings of Nanoparticle Tracking Analysis for each sample were captured using a SCMOS camera and analyzed by NTA v3.0 0068 software.

**Statistical analysis.** For comparison of two groups, two-tailed, unpaired $t$ test was used determined by GraphPad Prism 7 software. For comparison of three or more groups, one-way analysis of variance with Dunnett's multiple comparisons test was determined by GraphPad Prism 7 software. $P < 0.05$ was considered statistically significant.

**Reporting summary.** Further information on research design is available in the Nature Research Reporting Summary linked to this article.

## Data availability

Plasmid DNAs to generate NanoMEDIC have been deposited to Addgene with the identifier 138476-138482. The MS/MS data have been deposited to the jPOSTrepo with the identifier JPST000623. The NGS data have been deposited to the NCBI SRA with BioProject ID PRJNA560477. The source data underlying Fig. 1c, e–h, 2b, c, 3c–h, 4a–e, 5a, b, d, e, g, h, 6b, d, e, g, h, and f and Supplementary Figs. 1a–d, 2a, c, 3a–d, 4a–c, 5a–d, 6b–e, 7a, b, e, and f are provided as a Source Data file. All other relevant data are available from the authors upon reasonable request.

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

## Acknowledgements

We thank Dr. Yoshio Koyanagi, Dr. Kei Sato, Dr. Hirotaka Ebina, Dr. Kazuya Shimura, and Dr. Takayuki Miyazawa for plasmid constructs, advice and active discussions. We also thank Sho Hasegawa, Dr. Keisuke Okita, and Dr. Naohiro Hashimoto for providing 1383D2 iPS cells, 404C2 iPS cells, and Hu5 cells, respectively. We are grateful to Dr. Hidetoshi Sakurai for providing DMD patient iPS cells with an inducible MYOD1 transgene and to Jun Otomo and Ming-Ming Zhao for their advice on differentiation of skeletal muscle cells from iPSCs. We also thank Brad Calvin, Joseph Abad, and Dr. James Brady for providing the MaxCyte STX electroporator, processing assemblies, and advice on developing the suspension-based NanoMEDIC production system. We also thank Dr. Keiko Imamura and Dr. Haruhisa Inoue for providing the NEUROG2 cDNA and also advice on the cortical neuron differentiation. This research was supported in-part by the Practical Research Project for Rare/Intractable Diseases (to A.H. and J.K.) by AMED under Grant Number JP19ek0109293, T-CiRA Join Research program (to A.H.) by Takeda Pharmaceutical Company, the Core Center for iPS Cell Research (to A.H. and H.S.) by AMED under Grant Number JP19bm0104001, the Acceleration Program for Intractable Diseases Research utilizing disease-specific iPS cells (to H.S. and A.H.) by AMED under Grant Number JP19bm0804005, Intramural Research Grant (No. 28-6) for Neurological and Psychiatric Disorders of NCNP (to A.H. and H.S.), Grant-in-Aid for Young Scientists B by JSPS under Grant Number 17K15048 (to P.G.). Peter Gee was supported by the fellowship program for the Promotion of Internationalization of Research (CiRA, Kyoto University).

## Author contributions

P.G., J.K., and A.H. designed the study concept. P.G., M.S.Y.L., N.Sa., T.I., L.F.Y., X.H.W., M.A.W., A.K., K.W., N.Sh., Y.O.A., and Y.O. performed molecular cloning and T7E1 analysis, and produced NanoMEDIC particles. H.X., K.A.I., and M.S.Y.L established EGxxFP SSA reporter cell lines. Y.Ma., H.H., N.I., Y.Mi., and H.S. performed in vivo experiments. H.X. and H.S. helped the design of *mdx* experiments. M.Y. generated the NOG-*mdx* mice. M.I. performed proteome experiments and analysis. Y.F. and T.N. performed EM analysis. P.G. and A.H. wrote the manuscript.

## Competing interests

P.G. and A.H. are inventors on a filed patent (PCT/JP2019/027708) based on the delivery system, virus-like particle, production, and use for genome editing described here. Y.M., H.H., and N.I. are employees of Takeda Pharmaceutical Company. A.H. is a principle investigator (without salary) of the T-CiRA program funded by Takeda Pharmaceutical Company. All other authors declare no competing interests.
