## [Peer Review file · Nature Communications]

Reviewers' Comments:

Reviewer #1:

Remarks to the Author:

The manuscript by Gee et al. describes the production of a delivery vehicle for macromolecules, specifically, Cas9 RNPs for genome modification. They describe the development, optimization and characterization of a delivery system that builds off existing technologies to create a novel system cleverly termed "NanoMEDIC". They address important and relevant issues to the field of CRISPR technology such as off-target effects, cell viability, and transient Cas9 RNP expression. They use their system for both in vitro and in vivo models of Duchenne muscular dystrophy to perform exon skipping in the dystrophin gene locus. Overall, the paper is well written and presented in a logical, straightforward manner. Statistics appear correct although the "n" values are not always clearly presented or defined (consider a supplemental table). There are few comments and concerns listed below:

- For other microvesicle-based delivery systems alluded to in the manuscript, characterization of the size and concentrations of particles are made. For comparison, the size distribution and concentration should be determined.
- Similar to the above point, how are the concentrations determined? The concentrations of NanoMEDIC should be clarified. An example is Figure 3G. Both the text and figure legends say "increasing concentrations". Within the Methods section "NanoMEDIC inoculation" it states that 10ul of 80-fold concentrated particles were used. This promising technology will be limited unless there is a way to standardize use among research applications.
- Duchenne muscular dystrophy is used as the disease to demonstrate potential utility of the NanoMEDIC system, a bit more information on the molecular pathogenesis of this disease would be beneficial. For example, why is exon skipping of exon 45 a meaningful target by CRISPR/Cas9? There is also no demonstration of phenotypic improvement as a consequence of the achieved exon skipping.
- In NanoMEDIC delivery in vivo, the second paragraph states that rats were injected intravenously, however, not data was shown using intravenous injection. These data should be shown.
- In Supplemental Figure 2, a comparison of NanoMEDIC to other EV delivery systems is made, however, there is little information on how much effort went into optimizing each of those methods. Were they obtained from experts that produced them? There is clearly potential for bias here. A thorough side by side comparison with extensive quality control, comparable concentrations of particles, etc. should be made otherwise the authors should simply present how their system performs, compare it to published efficiencies of other systems and remove this section.

Reviewer #2:

Remarks to the Author:

In this manuscript Gee et al describe a delivery method for Cas9 and sgRNA based on non-viral vesicles. The authors exploit FKBP12-FRB dimerization properties following AP21967 (Rapamycin) treatment to induce incorporation of Cas9 in vesicles. The vesicles carry VSV-G to induce fusion with target cells. The sgRNA are transcribed from an HIV derived LTR promoter and carry a Psi signal for interaction with the HIV-Gag domain fused to FKBP12.

To test the efficacy of the derived vesicle, NanoMEDIC, the authors uses reporter systems (GFP) a DMD reporter construct for exon skipping readout and a luciferase construct carrying the DMD exon 45. Finally, NanoMEDIC are tested in mice carrying the Luciferase DMD exon 45 construct.

NanoMEDIC might be potentially a step forward for Cas9 delivery and application in DMD gene therapy, however further experimental work is needed to reinforce the solidity the tool. Several aspects of the manuscript require clarifications.

Major comments:

The majority of the NanoMEDIC testing was performed in reporter sequences, either integrated or not. Nevertheless, a rigorous estimation of the cleavage efficacy and off target activity require a broader editing analysis of endogenous loci.

The few loci analyzed produce controversial results: with the EMX comment NanoMEDIC lose activity with respect to other loci, why?

The efficacy of NanoMEDIC tested using the reporter DMD construct in animal models might over-estimate the effect of this tool for in vivo application targeting the endogenous DMD locus. Testing in DMD locus in animal model is necessary to draw conclusion on the efficacy of this tool for DMD treatment

It is not clear the requirement of the HIV TAR coupled with Tat. How Tat RNA elongation would be beneficial to sgRNA transcription/incorporation? The use of these viral elements would generate safety concern for its use in vivo

Sequence editing analysis is necessary to evaluate the indels generated after injection in mice. The luciferase assay used by the authors is not very sensitive in terms of spatial resolution.

Minor

- the off-target analysis should be expanded in particular towards the DMD locus

- The data reported on HIV protease activity is out of focus (Lines 464-476) and does not add relevant information to the NanoMEDIC reported. Data set can be removed or at best moved to supplementary explaining the rationale for this analysis.

- fig. 1A: why is there a myristoylation signal at the N-term of Cas9?

In addition, since myristoylation tethers Cas9 to the membrane, does this contribute to Cas9 incorporation in vesicles?

- In fig. 2B it seems that there efficiency is lost following expression using the ribozyme system (compare U6 and deltaPSI plus Tat). The efficiency of ribozymes excision of the gRNA from the viral construct should be tested.

- line 118: "to make" repeated twice

-line 685: LentiSLICES is not an "integration deficient" lentiviral system. There is no proof of linearization of the vector before integration.

- line 89: technically VESiCas are vesicles in which Cas9 is passively incorporated and not fused to Gag

- in fig. 1E: the difference in vesicles activity might not necessarily explained with different Cas9 incorporation unless Cas9 activity fused at the N- or C- terminus is evaluated. Indeed, results in fig. 1E might be due to different Cas9 activity following to its fusion rather than to incorporation. Cleavages activity of Cas9 fusions together with western blot analysis might clarify these results.

- fig. S1A: there is a lot of free VSV-G. Did the authors co-transfect also VSV-G wt together with the one fused to the dimerization domain? The levels of the EGFP fusion are much lower than those of the other fusions: is this an effect on incorporation efficiency?

-line 455-457: awkward. Rephrase.

- line 615: the intravenous inoculated mice data are not found in the manuscript. Only intramuscular reported

- the authors claim that that budding is due to Gag, nevertheless literature reports that VSV-G induces allows vesiculation. To test the contribute of Gag vesiculation in NanoMEDIC NO-vsvg control testing Cas9 in supernatant is required. Indeed, considering the VSV-G induction of vesicles formation reported in literature the NO-Gag control in fig S2B are quite surprising. Please comment.

- line 673: the dimerization system used by the authors cannot be reported as novel since already used (see Gesicle)

- consistency of data: compare fig. 2c last bar with 2 d upper panel last bar upper panel , same for the same for 3h and 4c for deletion percentage . Quantification of Cas9 applied should be reported .

- 4B does not report percentages of deletion

- line 484: reference to Fig.2D is missing.

- Fig. 4a: it is awkward that the % of indels generated by a single sgRNA (DMD1 or DMD23) is similar to the indels generated by the sgRNA in combination since the cleavages generating the deletion should not be detected by the same PCR analysis.

Reviewer #3:

Remarks to the Author:

Major point

In figure 6, what percentage of the reporter luciferase gene is deleted of exon 45 ? This information could be obtained by ddPCR of the DNA extracted form the muscle. As currently reported, this may be a very low percentage, yet this is important to access the value of this technology for a potential treatment.

Minor points

In figure 3E, why is there a X over the arrow for sgRNA DMD23 ? Is that sgRNA DMD1 or sgRNA DMD23 as in figure 3F?

In both Figure 3E and 3F why are the results 0 for sgRNA DMD23?

For figure 6C, the injection of NanoMEDIC is intra-muscular rather than inter-muscular as indicated on the figure.

Reviewers' comments: in black, our comments in blue.

Reviewer #1 (Remarks to the Author):

The manuscript by Gee et al. describes the production of a delivery vehicle for macromolecules, specifically, Cas9 RNPs for genome modification. They describe the development, optimization and characterization of a delivery system that builds off existing technologies to create a novel system cleverly termed "NanoMEDIC". They address important and relevant issues to the field of CRISPR technology such as off-target effects, cell viability, and transient Cas9 RNP expression. They use their system for both in vitro and in vivo models of Duchenne muscular dystrophy to perform exon skipping in the dystrophin gene locus. Overall, the paper is well written and presented in a logical, straightforward manner. Statistics appear correct although the "n" values are not always clearly presented or defined (consider a supplemental table). There are few comments and concerns listed below:

We would like to thank the reviewer for expressing interest in our study and for the constructive comments. To address the reviewer's concerns, we have performed additional experiments and included the definition of "n". Please find our point-by-point response to the reviewer's comments listed below.

- For other microvesicle-based delivery systems alluded to in the manuscript, characterization of the size and concentrations of particles are made. For comparison, the size distribution and concentration should be determined.

We would like to thank the reviewer for raising this important point. We have carried out further experiments to characterize NanoMEDIC. We purified NanoMEDIC by affinity column purification using a monolith quaternary amine column, and eluted samples by high NaCl solution. We found two populations of NanoMEDIC eluted at 0.65 M NaCl and 1 M NaCl, respectively. By Western blot analysis, we found the 0.65M eluted NanoMEDIC had less Cas9 protein than the 1 M NaCl eluted particles (Figure S4A). We next characterized NanoMEDIC by NanoSight to determine particle number and diameter, as well as by electron microscopy to observe the structures of the particles. We found that the 0.65 M eluted particles were slightly smaller in average diameter than 1 M eluted NanoMEDIC (Figure S4B-C). The purified NanoMEDIC particles were also subjected to an *in vitro* cleavage assay and the amount of active RNP complex was determined using a recombinant Cas9 protein control. In line with Western blot results, higher levels RNP correlated with higher in vitro cleavage activity of 1 M NaCl eluted NanoMEDIC samples. We also performed proteomics analysis of NanoMEDIC to get a better idea of the proteins being incorporated (Figure S2C and Supplemental Table 1).

- Similar to the above point, how are the concentrations determined? The concentrations of NanoMEDIC should be clarified. An example is Figure 3G. Both the text and figure legends say “increasing concentrations”. Within the Methods section “NanoMEDIC inoculation” it states that 10ul of 80-fold concentrated particles were used. This promising technology will be limited unless there is a way to standardize use among research applications.

For the optimization and development of NanoMEDIC, including protein incorporation and sgRNA incorporation, we generated various EVs in parallel and compared their activities by inoculating an equal volume of concentrated EVs on the same number of cells (Fig 1 and 2). However, for latter experiments, we quantified the amount of active RNP complexes per volume of concentrated EVs by an *in vitro* cleavage assay. Briefly, we lysed NanoMEDIC to release SpCas9 RNP and then added a dsDNA template (PCR product) to evaluate the cleavage activity by quantifying the cleaved DNA bands by TapeStation analysis. The cleavage efficiencies obtained were compared with those of a recombinant SpCas9 protein standard curve which was generated by incubating a range of recombinant SpCas9 protein with the DNA target sequence and *in vitro* transcribed sgRNA. Where applicable, we included the quantified RNP used for inoculation in the figure legend.

Other SpCas9 delivery papers have quantified the amount of SpCas9 protein by western or ELISA, however, this does not accurately represent the amount of active RNP complex as some SpCas9 protein is not loaded with sgRNA and therefore not functional.

- Duchenne muscular dystrophy is used as the disease to demonstrate potential utility of the NanoMEDIC system, a bit more information on the molecular pathogenesis of this disease would be beneficial. For example, why is exon skipping of exon 45 a meaningful target by CRISPR/Cas9? There is also no demonstration of phenotypic improvement as a consequence of the achieved exon skipping.

Exon skipping strategies by CRISPR-Cas9 are well-studied for DMD to restore the open reading frame of dystrophin, as a large portion of the rod domain of the dystrophin protein can be deleted without completely impairing the protein's function. This is evident in Becker's muscular dystrophy patients who have up naturally occurring large deletions in the rod domain (up to 50%) yet have much milder disease symptoms and higher survival. CRISPR-Cas9 mediated exon skipping could induce permanent restoration of the dystrophin open reading frame to eventually cure the patient. We have provided further explanations in the introduction as to why exon skipping would be a useful strategy for DMD.

- In NanoMEDIC delivery *in vivo*, the second paragraph states that rats were injected intravenously, however, not data was shown using intravenous injection. These data should be shown.

This was a mistake in the text and we have removed the intravenous injection sentence from the second paragraph.

- In Supplemental Figure 2, a comparison of NanoMEDIC to other EV delivery systems is made, however, there is little information on how much effort went into optimizing each of those methods. Were they obtained from experts that produced them? There is clearly potential for bias here. A thorough side by side comparison with extensive quality control, comparable concentrations of particles, etc. should be made otherwise the authors should simply present how their system performs, compare it to published efficiencies of other systems and remove this section.

The comparison with the other EV delivery systems was performed using DNA plasmids that were constructed in our laboratory and not by the groups who published them, as the constructs are not available. All EVs were produced by transfecting comparable DNA plasmid amounts and harvested in parallel. The same volume of EVs were inoculated onto reporter cells for the side-by-side comparison. However, as the reviewer mentioned, because the plasmids we generated may lead to different results than those published owing to coding sequence and expression vector differences as well as the transfection method used, we decided to remove the comparison of NanoMEDIC with the fused Gag-SpCas9 data.

For the comparison of our system with commercially available Gesicle system sold by Clontech, we generated the EVs according to the manufacturer's instructions and with their reagents, in parallel with NanoMEDIC and inoculated the same volume of EVs onto reporter cells. Therefore, we feel this comparison is fair and have left the data in Supplemental Figure 3D.

Reviewer #2 (Remarks to the Author):

In this manuscript Gee et al describe a delivery method for Cas9 and sgRNA based on non-viral vesicles. The authors exploit FKBP12-FRB dimerization properties following AP21967 (Rapamycin) treatment to induce incorporation of Cas9 in vesicles. The vesicles carry VSV-G to induce fusion with target cells. The sgRNA are transcribed from an HIV derived LTR promoter and carry a Psi signal for interaction with the HIV-Gag domain fused to FKBP12. To test the efficacy of the derived vesicle, NanoMEDIC, the authors uses reporter systems (GFP) a DMD reporter construct for exon skipping readout and a luciferase construct carrying the DMD exon 45. Finally, NanoMEDIC are tested in mice carrying the Luciferase DMD exon 45 construct. NanoMEDIC might be potentially a step forward for Cas9 delivery and application in DMD gene therapy, however further experimental work is needed to reinforce the solidity the tool. Several aspects of the manuscript require clarifications.

We would like to thank the reviewer for expressing interest in our study and for thoroughly evaluating our manuscript. To address the reviewer's concerns, we have performed additional experiments. Please find our point-by-point response to the reviewer's comments listed below.

Major comments:

The majority of the NanoMEDIC testing was performed in reporter sequences,

either integrated or not. Nevertheless, a rigorous estimation of the cleavage efficacy and off target activity require a broader editing analysis of endogenous loci.

For the development and optimization of the NanoMEDIC system, reporter cells were predominantly used for screening purposes and optimization of the fusion proteins, and sgRNA packaging. However, we showed that highly efficient endogenous gDNA cleavage was achieved in iPS cells (Fig. 3). We also showed VEGFA, CCR5 and SAMHD1 endogenous loci could be targeted by NanoMEDIC in HEK293T cells, and T cells as well as in cortical neurons differentiated from iPS cells, respectively (Fig. S5).

Nonetheless, to clarify the utility of NanoMEDIC for inducing mutations in multiple loci in the same cell line, we newly performed the cleavage assays of 6 endogenous genomic loci in HEK293T cells: DMD1, DMD23, SAMHD1#1, SAMHD1#2, CCR5#1, and EMX1. We show that at all 6 loci, NanoMEDIC could be titrated in a dose dependent manner to achieve higher indel efficiency than plasmid DNA transfection (Figure S5D).

The few loci analyzed produce controversial results: with the EMX comment NanoMEDIC lose activity with respect to other loci, why?

In the above revised experiments, we determined the amount of active RNP in NanoMEDIC targeting 6 loci and found that NanoMEDIC could be titrated in HEK293T cells up to 500 ng of active RNP to induce comparable or higher mutation frequencies to those observed with DNA plasmid transfection. Although editing at some loci such as EMX1 and DMD23 only had maximal indel rates of approximately 20%, in general, indels were observed to be higher than 40% for 4 out of 6 loci tested (DMD1, SAMHD1#1, SAMHD1#2, CCR5 in Fig. S4E). The low activity at some loci is most likely attributed to the intrinsic sgRNA activity, as plasmid DNA transfection with the same sgRNAs also showed relatively lower activities than the other sgRNAs.

The efficacy of NanoMEDIC tested using the reporter DMD construct in animal models might over-estimate the effect of this tool for in vivo application targeting the endogenous DMD locus. Testing in DMD locus in animal model is necessary to draw conclusion on the efficacy of this tool for DMD treatment

Our optimized sgRNA against the human DMD sequence cannot be tested in a mouse context, hence we created the Luc exon skipping reporter mice. We validated the mRNA exon skipping and genomic DNA cleavage in the Luc reporter mice in new Fig.6E-G.

It is not clear the requirement of the HIV TAR coupled with Tat. How Tat RNA elongation would be beneficial to sgRNA transcription/incorporation? The use of these viral elements would generate safety concern for its use in vivo

Our data indicates that Tat and TAR would be beneficial for both transcription and incorporation of sgRNA as evident by the functional all-in-one NanoMEDIC produced in the presence of Tat. Tat has been previously reported to enhance full transcription of mRNA. Since our mRNA has highly complex secondary

structure (Psi, hammerhead ribozyme, sgRNA, and hepatitis delta virus) sequences, these features could lead to premature termination of RNA transcription by RNA polymerase. Functionally, we show that Tat enhances the incorporation of sgRNA into NanoMEDIC. Future studies will focus on the efficiency of sgRNA transcription using these elements.

Regarding safety concerns, as seen in the proteomics analysis of NanoMEDIC particles, the amount of stochastically packaged Tat protein is relatively low and its presence is transient, as the protein would be cleared and would likely not pose a toxicity risk.

Sequence editing analysis is necessary to evaluate the indels generated after injection in mice. The luciferase assay used by the authors is not very sensitive in terms of spatial resolution.

To examine the mutation pattern induced in the stably integrated reporter sequence by NanoMEDIC, we performed MiSeq deep sequencing analysis and analyzed cleavage activity in three Luc reporter mice to indicate *in vivo* genome editing activity. We found the deletions within the region of the dystrophin gene targeted by sgRNA DMD#1 and #23 (Fig. 6G), with deletion sizes up to 130 bp (Fig. 6H). Furthermore, exon skipping by RT-PCR at Day 189 in three mice showed an average skipping rate of 6.9%.

Minor

- the off-target analysis should be expanded in particular towards the DMD locus

In the past, we have performed extensive off-target analysis of sgRNAs targeting DMD locus in DMD patient-derived iPS cells (Li HL et al., Stem Cell Reports, 2015). However, because we have been unable to detect any off-target activity with these sgRNAs, we decided to validate the reduced off-target activity of NanoMEDIC containing two sgRNAs with widely reported promiscuous off-target activity (EMX1 and VEGFA) in Fig. 5.

- The data reported on HIV protease activity is out of focus (Lines 464-476) and does not add relevant information to the NanoMEDIC reported. Data set can be removed or at best moved to supplementary explaining the rationale for this analysis.

We have moved this data to Figure S2A.

- fig. 1A: why is there a myristoylation signal at the N-term of Cas9?
In addition, since myristoylation tethers Cas9 to the membrane, does this contribute to Cas9 incorporation in vesicles?

This was a mistake on the diagram and there is no myristoylation signal at the N-term of Cas9. We have removed this labeling from the schematic. Thank you for noting our error.

- In fig. 2B it seems that there efficiency is lost following expression using the ribozyme system (compare U6 and deltaPSI plus Tat). The efficiency of ribozymes excision of the gRNA from the viral construct should be tested.

The difference in efficiency may be explained by the RNA length and different expression promoters. The sgRNA from a U6 promoter is only 120 nt long versus >2000 bp for the ribozyme construct from an LTR (RNA pol II) promoter and AmCyan reporter sequence which is transcribed with the addition of Tat. We have consistently observed in previous experiments that U6 promoter driven sgRNA expression in NanoMEDIC producer cells leads to higher indel induction in “these producer cells” compared with sgRNA expressed with the ribozyme system (RGR-AmCyan; hammerhead ribozyme-sgRNA-HDV ribozyme-AmCyan) from an EF1a promoter lacking the psi element or LTR promoter with Tat. In fact, the LTR driven RGR with Tat had the lowest indel induction in the “producer cells”. Importantly though, the amount of “packaged”

functional sgRNA is higher when using the ribozyme system in combination with an LTR promoter containing Psi, and supplemented with Tat. Please see the graph data below:

While the results here show functionality of the ribozyme system in conjugation with Psi and Tat mediated expression, future studies will examine the efficiency of ribozyme excision.

- line 118: “to make” repeated twice

We have deleted the repeat.

-line 685: LentiSLICES is not an “integration deficient” lentiviral system. There is no proof of linearization of the vector before integration.

We have removed the “integration deficient” phrase.

- line 89: technically VESiCas are vesicles in which Cas9 is passively incorporated and not fused to Gag

We have clarified this point in the text to make it clear that Cas9 is being passively incorporated into EVs with VESiCas.

- in fig. 1E: the difference in vesicles activity might not necessarily explained with different Cas9 incorporation unless Cas9 activity fused at the N- or C- terminus is evaluated. Indeed, results in fig. 1E might be due to different Cas9 activity following to its fusion rather than to incorporation. Cleavages activity of Cas9 fusions together with western blot analysis might clarify these results.

Based on the reviewer's comments, we have performed additional experiments to compare the activities of the N-, C-, and N- plus C- terminus fused Cas9 constructs. As seen in Fig. S1B, the activity of all three constructs is comparable with wild type SpCas9 when they are transfected into HEK293T cells, though the N- and C- terminus fused SpCas9 has slightly lower activity owing to slightly lower expression by Western blot analysis. When we looked at the amount of FRB fused SpCas9 proteins in NanoMEDIC, we found that all three proteins had relatively similar levels of protein in the presence of AP21967 with the C-terminus FRB fused SpCas9 having the highest amount of protein (Fig. S1C). Therefore, the reason for the higher activity of the N-terminus FRB fused SpCas9 construct may be presumably due to more efficient release from the EV in the target cell. We have changed the text accordingly in light of these new findings.

- fig. S1A: there is a lot of free VSV-G. Did the authors co-transfect also VSV-G wt together with the one fused to the dimerization domain? The levels of the EGFP fusion are much lower than those of the other fusions: is this an effect on incorporation efficiency?

VSV-G WT was used to supplement the VSV-G fusion construct because we were worried that the VSV-G fusion would have reduced fusogenicity and entry activity in the target cell. We believe the EGFP fusion protein is incorporated less efficiently into EVs.

-line 455-457: awkward. Rephrase.

We revised the sentence as follows:

"The resulting NanoMEDIC particles were subsequently inoculated onto EGxxFP SSA reporter HEK293T cells, which did not stably express sgRNA DMD1, and flow cytometry was performed three days post inoculation."

- line 615: the intravenous inoculated mice data are not found in the manuscript. Only intramuscular reported

This was a mistake in the text and we have removed the intravenous injection sentence from the second paragraph.

- the authors claim that that budding is due to Gag, nevertheless literature reports that VSV-G induces allows vesiculation. To test the contribute of Gag vesiculation in NanoMEDIC NO-vsvg control testing Cas9 in supernatant is

required. Indeed, considering the VSV-G induction of vesicles formation reported in literature the NO-Gag control in fig S2B are quite surprising. Please comment.

As the reviewer mentioned, both VSV-G and Gag are capable of inducing budding. Thus, we do not completely exclude the contribution of VSV-G to the budding of the EVs. However, compared with VSV-G alone, FKBP12-Gag + VSV-G was more efficient at inducing EVs containing SpCas9 even in the absence of AP29167 (Fig. S1A). We have not tested the no-VSV-G control as a comparison because these resulting EVs would not have entry activity when inoculated onto cells. Therefore, we removed the wording of Gag “to actively induce extracellular vesicles” in order to avoid confusion with the fact that VSV-G can also induce EVs.

In Fig S2B, we did not observe delivery of all-in-one EVs containing SpCas9 and sgRNA by VSV-G alone when compared with NanoMEDIC at the volume of EVs we tested. The packaging of SpCas9 stochastically does occur but the amount of protein is low (as observed above). Moreover, sgRNA stochastic incorporation from a U6 promoter in producer cells is insufficient for being packaged into EVs as we found in our study and was reported in the VesiCas system. In their paper, the authors had to overexpress a bacterial RNA polymerase in the cytoplasm of EV producer cells in order to achieve packaging of sgRNA. Despite this, even in their system, they had to inoculate their EVs up to three times to obtain sufficient genome editing indicating that the potency of their all-in-one EVs was low. Therefore, the best explanation we can provide is that both protein and sgRNA together as an active complex are insufficiently packaged.

However, as Reviewer 1 pointed out, the comparison between the different EV production systems may not be accurately represented in our paper since we do not have their original materials (DNA plasmids) and cells used in the other published studies to produce the EVs. Thus, we have decided to remove Fig. S2B. We will leave the comparison between our dimerization system with the Clontech Gesicle system because the reagents are commercially available from the company and produced the EVs as described in the manufacturer’s protocol in parallel with NanoMEDIC.

- line 673: the dimerization system used by the authors cannot be reported as novel since already used (see Gesicle)

We have rephrased the sentence on the first paragraph of the Discussion.

- consistency of data: compare fig. 2c last bar with 2 d upper panel last bar upper panel, same for the same for 3h and 4c for deletion percentage. Quantification of Cas9 applied should be reported.

When we performed the initial experiments while optimizing our NanoMEDIC system in Fig. 2C and 3H, we did not have a defined system for measuring the amount of active RNP complexes. However, for latter experiments, we quantified the amount of active RNP complexes per volume of concentrated EVs by an *in vitro* cleavage assay. Briefly, we lysed NanoMEDIC to release SpCas9 RNP and then added a dsDNA template (PCR product) to evaluate the cleavage activity by quantifying the cleaved DNA bands by TapeStation analysis. The cleavage

efficiencies obtained were compared with those of a recombinant SpCas9 protein standard curve which was generated by incubating a range of recombinant SpCas9 protein with the DNA target sequence and *in vitro* transcribed sgRNA. The new data is presented in Fig. S 5D.

Notably, other SpCas9 delivery papers have quantified the amount of SpCas9 protein by western or ELISA, however, this does not accurately represent the amount of active RNP complex as some SpCas9 protein is not loaded with sgRNA and therefore not functional.

- 4B does not report percentages of deletion

The percentages have been added to Figure 4B.

- line 484: reference to Fig.2D is missing.

The reference has been added.

- Fig. 4a: it is awkward that the % of indels generated by a single sgRNA (DMD1 or DMD23) is similar to the indels generated by the sgRNA in combination since the cleavages generating the deletion should not be detected by the same PCR analysis.

The indel% at the exon 45 splicing acceptor and splicing donor sites were determined using unique primers which amplified each targeted site independently in Fig.4A. When we assessed both sgRNA targeting sites by a single pair of primers in Fig.6F&G, the majority of deletions accrued individually, rather than large deletion spanning the both sites, which might explain the similar cleavage activities between single and multiplexed sgRNAs.

Reviewer #3 (Remarks to the Author):

Major point

In figure 6, what percentage of the reporter luciferase gene is deleted of exon 45? This information could be obtained by ddPCR of the DNA extracted from the muscle. As currently reported, this may be a very low percentage, yet this is important to assess the value of this technology for a potential treatment.

To examine the mutation pattern induced in the stably integrated reporter sequence by NanoMEDIC, we performed MiSeq analysis and analyzed cleavage activity in three Luc reporter mice to indicate *in vivo* activity. We found the deletions within the region of the dystrophin gene targeted by sgRNA DMD#1 and #23 (Fig. 6F,G), with deletion efficiency of 2%. Furthermore, exon skipping by RT-PCR at Day 189 in three mice showed an average skipping rate of 6.9%.

Minor points

In figure 3E, why is there a X over the arrow for sgRNA DMD23 ? Is that sgRNA DMD1 or sgRNA DMD23 as in figure 3F?

In both Figure 3E and 3F why are the results 0 for sgRNA DMD23?

We apologize for the confusion. There is an X over the arrow for sgRNA DMD23 because the targeting sequence that was inserted between the GFP coding region does not contain the DMD23 targeting sequence. Therefore, only DMD1 targeting NanoMEDIC should result in GFP expression upon single strand annealing after cleavage. DMD23 sgRNA does not target the reporter and would result in no activity.

The following sentence was added to the Figure 3E-F legend to avoid confusion: "Only the DMD1 targeting sequence is contained in the reporter, not the DMD23 targeting sequence, which is indicated with an X."

For figure 6C, the injection of NanoMEDIC is intra-muscular rather than inter-muscular as indicated on the figure.

We have revised the sentence to indicate intra-muscular and not inter-muscular.

Reviewers' Comments:

Reviewer #1:

Remarks to the Author:

The authors have addressed the majority of the issues raised in the initial review and the manuscript is improved. The issue of comparison with other technologies is the remaining concern. The authors remove the comparison with Gag-SpCas9 but left the vesicle comparison. They claim it is fair based on using protocol and then used "same volume." In this comparison, they have used NanoMedic which were quantified and qualified in numerous ways to be optimal but then did no such analysis with vesicles. Additional QC should be done with vesicles such as Nanocyte, western blot and active RNP per volume. Perhaps the other methods require more cells or material but have higher specific activity. The attempt to compare needs more rigor or should be removed.

Reviewer #2:

Remarks to the Author:

The authors have done quite a good job in improving the manuscript. However there are still important issues to be addressed to sustain the final conclusion on the *vivo* efficacy of NanoMEDIC.

In the new version of the manuscript the authors analysed the editing profile (deletion size and position in fig. 6f-g). However, the authors omit to report the editing efficiency. This can be easily calculated from the sequences reads reported in fig. 6f by using the total number of reads obtained in the experiments. The editing efficiency is highly informative.

Moreover, to have a more realistic information on the editing efficacy of NanoMEDIC *in vivo* the authors should test the endogenous mouse dystrophin locus. Indeed, it is well known that exogenous sequences are more easily edited than endogenous loci. Finally, since so far Cas editing of dystrophin in mice has been obtained with AAV (last report from Gersbach lab, Nat Med) with quite limited efficacy it would be important to compare it with NanoMEDIC efficacy.

Minor remarks:

-Figures 6E-F are not described in the text. In the main text authors describe data up to Fig. 6D. - The use of Tat has a number of related safety issues. Indeed, vesicles formed from Tat expressing cells may carry tat protein as well as mRNA expressing Tat and Tat expressing DNA. The authors should at least comment these aspects of potential issues.

Reviewer #3:

Remarks to the Author:

Thank you for the revised version of the manuscript.

Point-by-Point Response Letter for Reviewers' comments (NCOMMS-19-07887)

Reviewer #1 (Remarks to the Author):

The authors have addressed the majority of the issues raised in the initial review and the manuscript is improved. The issue of comparison with other technologies is the remaining concern. The authors remove the comparison with Gag-SpCas9 but left the gesicle comparison. They claim it is fair based on using protocol and then used "same volume." In this comparison, they have used NanoMedic which were quantified and qualified in numerous ways to be optimal but then did no such analysis with gesicles. Additional QC should be done with gesicles such as Nanocyte, western blot and active RNP per volume. Perhaps the other methods require more cells or material but have higher specific activity. The attempt to compare needs more rigor or should be removed.

We would like to thank the Reviewer for acknowledging the improvement of our revised manuscript.

Regarding the comparison with the Clontech Gesicle system, this commercial product comes as a kit with predefined reagents, such as transfection reagent, pre-mixed plasmid DNA constructs with undisclosed sequence map and mixture ratio. Therefore, we were unable to further optimize this kit as we can with our own plasmids. We followed the manufacturer's detailed protocol regarding cell density, heterodimerizer concentration, and timing of transfection and harvest of the particles. In addition, inoculation of Gesicle is recommended to perform in volume in the protocol (i.e. Add 30 μ l of the thawed Cas9/sgRNA gesicles from Step 1 to each well). Therefore, in Supplementary Figure 3D, we attempted to produce our NanoMEDIC particles in parallel with the Clontech Gesicle system in the same production scale (10 cm dish) and inoculate the same volume (10 μ L) of concentrated extracellular vesicle products.

We believe the main reason for the improved efficiency of our system is due to the Gag based dimerization being superior to incorporating Cas9 protein compared with other membrane anchoring proteins (i.e. CherryPicker in Gesicle system).

To demonstrate this with the Clontech system, we performed preliminary experiments to quantify Cas9 protein by Western blotting (before we optimized sgRNA packaging), and we found that although the expression of SpCas9 protein using the Clontech system is higher than our NanoMEDIC system in EV producer cells, the amount of packaged SpCas9 protein is much higher in released extracellular vesicles (EVs) by NanoMEDIC than Gesicle (right data). Thus, we believe the difference in the genome editing efficiency in Fig. S3D is not due to an issue with transfection nor expression of the components in the producer cells, but rather improved

Note: Unmarked lanes are unrelated NanoMEDIC samples.

production of the NanoMEDIC particles containing higher amounts of SpCas9 protein.

Further characterization of the Clontech Gesicle system (i.e. counting particle number or active RNP complex) would not resolve the issue of Cas9 protein incorporation, and if we tried to normalize the inoculation amount, we have to produce Gesicles several times larger scale, in which the kit costs over 1,000 USD for production in ten 10 cm dishes. We hope the reviewer understands that we have done extensive preliminary work with the Clontech system alongside with our NanoMEDIC before we optimized sgRNA loading conditions.

Our intention is to show our NanoMEDIC's performance alongside with publicly available commercial kit as is. Therefore, we believe characterization or optimization of the Clontech Gesicle system is beyond the scope of our paper.

Reviewer #2 (Remarks to the Author):

The authors have done quite a good job in improving the manuscript. However there are still important issues to be addressed to sustain the final conclusion on the vivo efficacy of NanoMEDIC.

In the new version of the manuscript the authors analysed the editing profile (deletion size and position in fig. 6f-g). However, the authors omit to report the editing efficiency. This can be easily calculated from the sequences reads reported in fig. 6f by using the total number of reads obtained in the experiments. The editing efficiency is highly informative.

We would like to thank the Reviewer for evaluating our manuscript and giving us constructive feedback.

We have calculated the editing efficiency at the targeted site for the % of DNA deletion, which is included in Fig. 6H-I, and corresponds to approximately 7%, similar to exon skipping activity previously reported in Fig. 6E.

Moreover, to have a more realistic information on the editing efficacy of NanoMEDIC in vivo the authors should test the endogenous mouse dystrophin locus. Indeed, it is well known that exogenous sequences are more easily edited than endogenous loci.

Regarding the editing of endogenous loci, all of our sgRNAs (DMD #1, #23, EMX1, SAMHD1 #1, #2, and CCR5) are designed to target human genomic sequences, therefore we cannot assess their editing efficiencies in a mouse context. As the Reviewer pointed out, the genome editing efficiency varies among sgRNA targeting sites or cell types, regardless of plasmid transfection or NanoMEDIC inoculation (Fig. S5D), rather depending on the sgRNA target, due to various factors (i.e. open transgene locus V.S. unexpressed endogenous gene locus). To properly show the variation of indel efficiencies in a mouse muscle tissue context, the addition of one or two more sgRNAs would not be enough, and such a comprehensive mouse experiment is beyond our lab's capacity. Instead, we added the following sentences below in the results section to clarify the limitation for readers about the potential

overestimation of the indel efficiency in our mouse experiment by targeting the transgene locus.

“It is worth noting the editing efficiency may be overestimated as we targeted a reporter gene locus, instead of the endogenous mouse *dystrophin* locus. Nonetheless, our data clearly addresses the ability of NanoMEDIC to be functionally delivered *in vivo* for exon skipping and genomic DNA targeting.”

Finally, since so far Cas editing of dystrophin in mice has been obtained with AAV (last report from Gersbach lab, Nat Med) with quite limited efficacy it would be important to compare it with NanoMEDIC efficacy.

As for the comparison with the AAV vector delivery system, it is very difficult to properly compare side-by-side, as various laboratories use different experimental conditions, such as Cas9 orthologue, sgRNA target site, AAV serotype, promoter type, viral titer, injection site, timing of tissue correction, and indel detection method. For your information, we tried to summarize the indel percentage of mouse genomic locus in the previous reports below. Considering the transient protein delivery nature of our NanoMEDIC, our *in vivo* editing efficiency (~7%) is comparable.

Model animal	Delivery	Genomic Indel%	Reference
mdx mouse	AAV9 SaCas9	% not shown (Fig. 2A)	Tabebordbar M. et al., Science, 2016
mdx mouse	AAV8 SaCas9	~2% (Fig. 1C)	Nelson CE. et al., Science, 2016
mdx mouse	AAV9 SpCas9	% not shown (Fig. S5)	Long C. et al., Science, 2016
mdx mouse	Adeno SpCas9	% not shown (Fig. 1D)	Xu L. et al., Mol. Ther, 2016
mdx ^{4cv} mouse	AAV6 Sp or SaCas9	~8% (Fig. 2A)	Bengtsson N. E. et al., Nat Commun, 2017
mdx50 mouse	AAV9 SpCas9	~20% (Fig. S7)	Amoasii L et al., Sci Trans Med, 2017
Ex23 indel mouse	AAV2/9 CjCas9	~8% (Fig.2C)	Koo T et al., Mol Ther, 2018
Δ 52hDMD/ mdx mouse	AAV9 SaCas9	% not shown (Fig.5)	Duchêne BL et al., Mol Ther, 2018
mdx44 mouse	AAV9 SpCas9	~15% (Fig. S3C, S4C)	Min YL et al., Sci Advances, 2019
mdx mouse	AAV8 SaCas9	~8% (Fig. 1C)	Nelson CE et al., Nat Med, 2019

Minor remarks:

-Figures 6E-F are not described in the text. In the main text authors describe data up to Fig. 6D.

Thank you for bringing this error to our attention. We have added the Figure references to the main text.

-The use of Tat has a number of related safety issues. Indeed, vesicles formed from Tat expressing cells may carry tat protein as well as mRNA expressing Tat and Tat expressing DNA. The authors should at least comment these aspects of potential issues.

Also, regarding the issue of Tat, we have added a paragraph in the discussion to address potential risks of Tat contamination in the EV preps.

“It is worth noting that our NanoMEDIC system utilizes HIV-1 Tat to drive the expression of sgRNA in producer cells and there is a risk of nonspecific incorporation of Tat into EVs as a protein, mRNA or possibly DNA. This poses a potential toxicity risk in recipient cells. Our proteomics data showed that while Tat protein levels were low in comparison to other host proteins incorporated into NanoMEDIC (Supplementary Table 1) and we did not observe any difference in cell proliferation of NanoMEDIC inoculated HEK293T recipient cells (Fig. 5B), we cannot completely rule out the possibility that Tat may have an effect on recipient cells.”

Reviewer #3 (Remarks to the Author):

Thank you for the revised version of the manuscript.

We would like to thank the Reviewer for the favorable feedback regarding our revised manuscript.

Reviewers' Comments:

Reviewer #1:

Remarks to the Author:

Were the EVs isolated using the same procedure for both gesicles and NanoMEDIC? If it was just an issue of the dimerization being superior, why so much less VSVG in the gesicles at the "packaged level"? It actually appears that the ratio of SpCas9 to VSVG is higher in the gesicles (i.e. maybe a higher specific activity per VSVG particle). Further characterization would be useful towards understanding the observed differences which "efficiency of dimerization" appears inconsistent with the results. If the authors are going to claim superiority of their method to another, there needs to be a thorough comparison especially if they are trying to commercialize the NanoMEDIC and use this article as a source of superiority. Although costly, resources could be diverted to providing the side by side comparison of the two methods. Otherwise, remove the comparison. The paper is solid without it.

Reviewer #2:

Remarks to the Author:

Since the main outcome of this work is the in vivo efficacy of NanoMEDIC as a potential therapeutic approach for exon skipping, I believe that the in vivo editing of an artificial locus cannot provide solid information on the efficacy of the reported system in mice. Indeed around 7% editing of the exogenous locus reported in the revised version, might be very far from the editing efficacy of an endogenous locus. The table reported by the authors in the rebuttal letter might be used as benchmark to test NanoMEDIC towards the endogenous mouse dystrophin locus. As emerging from the table SpCas9 efficacy in the mdx model is not very heterogeneous (excluding the other orthologues SaCas and CjCas). The homogenous range of activity in different studies (reported in the table) demonstrate that testing the dystrophin locus would give a real indication of the NanoMEDIC system with respect to conventional AAV delivery.

Point-by-point Response Letter: NCOMMS-19-07887B

Dec 31, 2019

Reviewer #1

We really appreciate the Reviewer for acknowledging that our manuscript is strong enough without the Gesicle comparison, and also your concerns about suboptimal Gesicl production condition. Please find our responses in-lines below.

Were the EVs isolated using the same procedure for both gesicles and NanoMEDIC?

Both the Gesicle and NanoMEDIC EVs were produced at the same time, same scale of HEK293T cells, and concentrated into a same volume by centrifugation in parallel. Transfection reagent is probably different from ours, because the Gesicle packaging plasmids and transfection reagents are pre-mixed in the kit.

If it was just an issue of the dimerization being superior, why so much less VSVG in the gesicles at the “packaged level”? It actually appears that the ratio of SpCas9 to VSVG is higher in the gesicles (i.e. maybe a higher specific activity per VSVG particle).

The difference of the VSV-G protein amount is probably due to the difference of VSV-G expression plasmid (undisclosed by the supplier) used in the Gesicle kit. Hence, we feel normalization of genome editing activity by the VSV-G amount is not appropriate nor a practical way to compare the two systems.

Further characterization would be useful towards understanding the observed differences which “efficiency of dimerization” appears inconsistent with the results. If the authors are going to claim superiority of their method to another, there needs to be a thorough comparison especially if they are trying to commercialize the NanoMEDIC and use this article as a source of superiority. Although costly, resources could be diverted to providing the side by side comparison of the two methods. Otherwise, remove the comparison. The paper is solid without it.

With the Gesicle kit, we have no control over the plasmid amount or ratios, because the Clontech Gesicle DNA plasmids are pre-mixed and lyophilized. Therefore, it is difficult to pin point precise differences between both systems, and to optimize the Gesicle system. In addition, we have developed our NanoMEDIC system for purely scientific advancements and no intension of making profit out of this. Indeed, we are depositing our plasmid DNA vectors to Addgene (ID: 138476-138482) to be openly available for scientists in the field.

Surely, we have no objection to remove the Gesicle comparison data as characterization and optimization of the commercially available kit is out of our scope, we were requested from the Journal Editor to keep the comparison data in the supplementary figure as a refence for the readers. To reflect your comments that there are challenges in drawing direct parallels and conclusions between the two

systems, we modified the description of the Discussion section to clarify these points on Page 23.

Reviewer #2:

Since the main outcome of this work is the in vivo efficacy of NanoMEDIC as a potential therapeutic approach for exon skipping, I believe that the in vivo editing of an artificial locus cannot provide solid information on the efficacy of the reported system in mice. Indeed around 7% editing of the exogenous locus reported in the revised version, might be very far from the editing efficacy of an endogenous locus. The table reported by the authors in the rebuttal letter might be used as benchmark to test NanoMEDIC towards the endogenous mouse dystrophin locus. As emerging from the table SpCas9 efficacy in the mdx model is not very heterogeneous (excluding the other orthologues SaCas and CjCas). The homogenous range of activity in different studies (reported in the table) demonstrate that testing the dystrophin locus would give a real indication of the NanoMEDIC system with respect to conventional AAV delivery.

We thank the reviewer for encouraging us to perform further in vivo mouse experiments. To assess the ability of our NanoMEDIC to target an endogenous gene locus, we have produced NanoMEDIC particles packaging two previously reported SpCas9-gRNAs that target exon 23 of the mouse dystrophin gene, one targets the point mutation site, and the other targets near the splicing donor site (Long C. et al., Science, 2016), to induce exon 23 skipping. Seven days after inoculation of the dual gRNA NanoMEDIC into TA muscle of *mdx* mice, genomic DNA was extracted from the TA muscle and analyzed by PCR and showed 1.1% of dual cleavage resulting in a 194 bp deletion (new Fig. 6G). Notably, this number does not include small indels at each sgRNA target site. Furthermore, when we examined mRNA, extracted from the TA muscle of the *mdx* mice, for exon 23 skipping efficiency by RT-PCR we could detect 1.5% exon skipping efficiency (new Fig. 6H).

While these efficiencies were rather low compared with our previous Luc-reporter mice, it might be due to the difference of targeting a reporter gene versus an endogenous gene locus, as the Reviewer suggested previously. We include these descriptions in the result section on Page 21.

Reviewers' Comments:

Reviewer #2:

Remarks to the Author:

The authors have addressed the main concern related to the NanoMEDIC efficacy in an endogenous locus. The manuscript is now reporting an important information and a notion of fair on the efficacy of NanoMEDIC tool depending on their application towards an endogenous locus or a transgene.